

# How Robust is Your System Resilience?

Mehran Homayounfar[1], Rachata Muneepeerakul[1], & John M. Anderies[2]

[1]Department of Agricultural and Biological Engineering, University of Florida, Gainesville, Florida, USA,
[2]School of Sustainability and School of Human Evolution and Social Change, Arizona State University, Tempe, Arizona, USA

*Correspondence to*: Mehran Homayounfar (homayounfar@ufl.edu)

**Abstract.** Robustness and resilience are concepts in systems thinking that have grown in importance and popularity. For many complex social-ecological systems, however, robustness and resilience are difficult to quantify and the connections and trade-offs between them difficult to study. Most studies have either focused on qualitative approaches to discuss their connections or considered only one of them under particular classes of disturbances. In this study, we present an analytical framework to address the linkage between robustness and resilience more systematically. Our analysis is based on a stylized dynamical model that operationalizes a widely used conceptual framework for social-ecological systems. The model enables us to rigorously delineate the boundaries of conditions under which the coupled system can be sustained in a long run, define robustness and resilience related to these boundaries, and consequently investigate their connections. The results reveal the tradeoffs between robustness and resilience. They also show how the nature of such tradeoffs varies with the choices of certain policies (e.g., taxation and investment in public infrastructure), internal stresses and external disturbances.

## 1. Introduction

The concepts of "resilience" and "robustness" have grown considerably in popularity as desirable properties for a wide range of systems. Terms like "resilient communities" and "robust cities" have been used more frequently in public discourse. Growing with that popularity is some confusion and potential misuse of the terms "robustness" and "resilience" due to imprecision, vagueness, and multiplicity of their definitions. Such lack of consistency and rigor hinders advances in our understanding of the interplay between these two important system properties.

Relatively speaking, robustness has been defined more consistently and rigorously—as it can be linked to a more familiar concept of sensitivity. For example, according to Carlson and Doyle, (2002), robustness in engineering systems refers to the maintenance of system performance either when subjected to external disturbances or internal uncertain parameters. In other words, in robust systems, performance is less sensitive to well-defined disturbances or stresses.

Robustness may very well be a desirable property of a system, but it seems to come with a price. Recent research shows that tuning a system to be robust against certain disturbance regimes almost always reduces system performance and likely increases its vulnerability to other disturbance regimes (Ostrom et al., 2007; Anderies et al., 2007; Bode, 1945; Csete and Doyle, 2002). Now, if resilience is also a desirable property of the same system, does it also come at



the expense of performance and robustness? Put it another way, is there a tradeoff among efficiency, robustness, and resilience? Such a tradeoff, if exists, is a crucial consideration for governing and/or managing coupled infrastructure systems[1] (CISs).

But resilience, as alluded to above, is trickier to define. According to Holling (1973), resilience refers to the amount of change or disruption required to shift the maintenance of a system along different sets of mutually reinforcing processes and structures. In other words, resilience can be thought of as how far the system is from certain thresholds or boundaries beyond which the system will undergo a regime shift or a quantitative change in system structure or identity. Holling (1996) categorized resilience into two types, engineering resilience, which refers to the ability of a system to return to steady state following a perturbation, and ecological resilience, which refers to the capacity of system to remain in particular stability domain in the face of perturbations. The latter category is used by many researchers to discuss resilience of social ecological systems (or more generally CISs) (Gunderson et al., 1995; Berkes and Folke, 1998; Carpenter et al., 1999a, 1999b; Scheffer et al., 2000; Anderies et al., 2002; Gunderson and Holling, 2002; Berkes et al., 2003; Walker et al., 2004; Carpenter and Brock, 2004; Janssen et al., 2004; Anderies, 2005; Folke et al., 2002; Anderies et al., 2006). The problem is that these CISs are complex and thus identifying thresholds and potential regime shifts associated with those thresholds is often difficult, if not impossible. In many cases, major aspects of resilience in SESs may not be directly observable and must be actualized indirectly via surrogate attributes (Carpenter et al., 2005; Kerner and Thomas, 2014). Recent significant advances have been made toward identifying early-warning signals that indicate for a wide class of systems whether the systems are approaching a critical threshold (Scheffer et al., 2009 and 2012). Still, there are gaps in our understanding of how indicators of resilience and robustness will behave in more complex situations. This lack of rigorous metric of resilience makes the investigation into their connections, interplay, and tradeoffs with robustness and performance impossible.

But these knowledge gaps need to be filled if one wishes to make advances in understanding the interplay between social dynamics and planetary boundaries. Given the magnitude of impacts that human activities have on pushing Earth systems toward their planetary boundaries, we need clearer understanding of how social and biophysical factors come together to define the nature of these boundaries. This paper is a step in that direction. In particular, we will build on recent work that mathematically operationalizes the Robustness of SES framework (Anderies et al., 2004) into a formal stylized dynamical model (Muneepeerakul and Anderies, 2017). We will exploit the relative simplicity of the model to rigorously define robustness and resilience of the coupled system. The modeled system will be subject to fluctuations in external drivers, which will affect the well-defined robustness and resilience, thereby enabling us to investigate the interplay and tradeoffs between these important properties, as well as how the nature of the interplay and tradeoffs are affected by policies implemented by social agents.

---

[1] The term *coupled infrastructure systems* (CISs) is introduced to generalize coupled natural-human systems and social-ecological systems; in this context, *infrastructure* is broadly defined to include human-made, social, and natural infrastructure.



## 2. Methods

Here we analyse a mathematical model developed by Muneepeerakul and Anderies, (2017) by subjecting the coupled system to uncertainty in ecological and social factors. The model captures the essential features of a system in which a group of agents shares infrastructure to produce valued flows. Such a system is the archetype of most, if not all of

human sociality: groups produce infrastructure that they cannot produce individually (security, defence, irrigation canals, roads, markets, financial systems, coordination mechanisms, etc.) that significantly increases productivity. The challenge is maintaining this shared infrastructure (e.g. decaying infrastructure is a major problem in the US at the time of writing (cite report of engineers)). The model allows for mathematical definitions of the boundaries of policy domain that result in a sustainable system in which both human-made and natural infrastructure can be

maintained over the long run. Based on these boundaries and uncertainty in the exogenous factors, we define metrics of resilience and robustness associated with each policy choice and investigate the tradeoff between them. The basic model presented by Muneepeerakul and Anderies, (2017) is described in the Appendix, which key variables and parameters are briefly explained below.

Here a policy is defined as a combination of taxation level $C$ and the proportion of tax revenue invested in

infrastructure maintenance $y$ that the public infrastructure providers (PIPs) decide to implement in the system. The infrastructure (e.g. canals) enable resource users (RUs) to produce valued goods from a natural resource. The two fluctuating exogenous factors are the replenishment rate of the natural resource $g$ and the wage $w$ that resource users (RUs) would earn from working outside the system—a combination of $g$ and $w$ defines a scenario. There are two boundaries that, once crossed, will cause the system will collapse. The first boundary is called PIP participation

constraint (PPC): when the PIPs must invest too much in maintaining the public infrastructure (exceeding the opportunity cost of $w_P$) and/or cannot retain enough revenue for themselves, they will abandon the system for another. The second boundary is the stability condition of the non-trivial equilibrium point (i.e., the "society" in which both PIPs (e.g. the state) and RUs (e.g. citizens) participate in the system and public infrastructure is sufficiently maintained in a long run). Together, these two boundaries delineate a set of policies ($C$-$y$ combinations) that correspond to

sustainable outcomes. The resilience metric to be developed below can be thought of as a metric of how far the system is from these boundaries. As the two exogenous factors defining scenarios, namely $g$ and $w$, fluctuate, the two boundaries and thus the resilience metric, too, fluctuate with them. How sensitive the resilience metric is to these fluctuating scenarios is used to define robustness. In the following sections, we define resilience and robustness more formally.

### 2.1. Resilience metric

Direct measurement of resilience in SES's is difficult because boundaries and thresholds that separate domains of dynamics for SES's are difficult to identify (Carpenter et al., 2005; Scheffer et al., 2009 and 2012). In this stylized model, however, such boundaries can be clearly identified, namely the stability condition (SC) and the PPC. We will now define resilience metrics based on these two boundaries. Here our goal is to develop resilience metrics that can



be meaningfully compared to one another. As such, we identify some desired properties that guide the definitions of these resilience metrics. First, they should be zero at their respective boundaries. Second, positive values indicate greater resilience of the system in a desirable state. These first two properties align with how resilience has been measured, i.e., the distance from the boundary of a basin of attraction (e.g., Anderies et al., 2002; S. R. Carpenter et al., 1999). Third, to facilitate the consideration of relative risks associated with different types of regime shifts that the system may be facing, the metrics should be comparable in magnitude. These properties guide us toward the following definitions of the resilience metrics.

We define the resilience of the system against abandonment by PIPs as follows:

$$R_{PPC} = (\pi_P/w_P) - 1,$$

(1)

where $\pi_P$ is the net revenue that PIPs collect and $w_P$ is the opportunity cost that they will earn if they choose to work with another system. Positive values of $R_{PPC}$ indicate that the system is resilient against being abandoned by PIPs, while negative values indicate that the system will eventually collapse due to the PIPs' abandonment.

Numerical analysis of the model indicates that the equilibrium becomes unstable when the following Routh-Horowitz condition is violated:

$$D - T(J_{1,1}J_{2,2} + J_{2,1}J_{1,2} + J_{2,3}J_{3,2} + J_{1,3}J_{3,1}) > 0,$$

(2)

where, $D$, $T$, and $J's$ are determinant, trace, and entries, respectively, of the Jacobian matrix of the dynamical system (Eqs. A1, A4, and A5) evaluated at the nontrivial equilibrium point (Eq. A6)—when such an equilibrium point exists. Following the guideline provided by the three desirable properties above, we rearrange terms in Eq. (2) and define the resilience of the system against instability as follows:

$$R_{stability} = \frac{D}{|T(J_{1,1}J_{2,2} + J_{2,1}J_{1,2} + J_{2,3}J_{3,2} + J_{1,3}J_{3,1})|} - 1.$$

(3)

This formulation is parallel to that of the first resilience metric (Eq. 1); it possesses the three properties: $R_{stability}$ of 0 indicates the boundary between stability and instability; positive $R_{stability}$ means the system at the equilibrium point is stable; and the magnitudes of $R_{stability}$ are comparable to those of $R_{PPC}$ (see Fig. 1). It is important to note that $R_{stability}$ is determined from the coupled dynamics of the coupled infrastructure system; this means that it has already integrated the dynamics of infrastructure, resource, and resource users (Eqs. A1, A4 and A5), making it a metric of the system, not of an individual component.

This allows us to meaningfully define the overall system resilience as the minimum between the two resilience metrics, namely:

$$R_{system} = \begin{cases} Min[R_{PPC}, R_{Stability}], & R_{PPC}, R_{stability} \geq 0 \\ 0, & otherwise \end{cases}.$$

(4)





Equation (4) implies that $R_{system}$ is positive only when the nontrivial equilibrium point (Eq. A6) exists and is stable; otherwise, the system is considered not resilient and denoted by $R_{system} = 0$. Given that the nontrivial equilibrium point exists and is stable, if the system is at a greater risk of being abandoned by the PIPs (and eventually collapsing), $R_{system} = R_{PPC}$; if the system is at a greater risk of becoming unstable (and eventually collapsing), $R_{system} = R_{stability}$. Figure 1 illustrates the relationships between $R_{PPC}$, $R_{stability}$, and $R_{system}$.

Figure 1: Resilience metrics for a specific scenario (a $g$-$w$ combination) inside the sustainable region in the policy space (i.e., $C$-$y$ plane): (a) $R_{PPC}$ contours; (b) $R_{Stability}$ contours; and (c) $R_{system}$ contours.

## 2.2. Robustness of resilience

As discussed earlier, robustness can be thought of as the opposite of sensitivity. One measure of sensitivity is variance. As such, variance of a given function under specific disturbance regimes can be used to indicate robustness of that function against those disturbance regimes (robustness of what to what). In this case, the system function of interest is the system resilience $R_{system}$. Therefore, we define the "robustness of resilience" by the variance of $R_{system}$ under a given disturbance regime (see below)—with lower variance indicating greater robustness. We will use $\sigma_{R-system}$ to denote this variance-based robustness metric.

In this study, we subject the modelled system to uncertainty in one natural factor and one social factor, namely, the natural replenishment rate of the resource $g$, and the payoff that a RU earns from working outside the system $w$. In particular, we assume that $g$ is uniformly distributed over the range [75, 125] and $w$ is uniformly distributed over the range [0.75, 1.75]. A scenario is defined as a combination of $g$ and $w$. For a given policy (a $C - y$ combination), we calculate $R_{system}$ for 10,000 scenarios (i.e., 10,000 $g - w$ combinations) (see Fig. 2). Then, from these 10,000 values of the resilience metric, $R_{system}$, we calculate the mean, $\mu_{R-system}$, and the standard deviation, $\sigma_{R-system}$: the mean $R_{system}$ is then used as the **resilience metric** of the coupled system with a given policy, and the standard deviation is used as the **metric for robustness of resilience** (lower variance/standard deviation mean greater robustness).

Figure 2: Variation of $R_{system}$ of a CIS with a fixed policy $(C, y)$ over 10,000 scenarios associated with a disturbance regime characterized by $\{g \in [75,125], w \in [0.75,1.25]\}$: (a) $R_{system}$ surface and (b) $R_{system}$ contours. The values of $R_{system}$ are used to calculate the mean, $\mu_{Rsystem}$, and the standard deviation, $\sigma_{Rsystem}$, of the variation of $R_{system}$. In this particular case, the resilience does not change much when $g$ is greater than about 100, but becomes more sensitive to both $g$ and $w$ when $g$ is lower than 100.

## 3. Results

The surfaces and contours of the system resilience metric, $\mu_{Rsystem}$, and associated with different policies $(C - y)$ over the policy space are shown in Figs. (3a and b), respectively. The policies with sustainable outcomes are located in the middle of the policy space, with $\mu_{Rsystem}$ peaking in the center and declining as policies become more extreme in either direction. Our analysis also shows that $\mu_{Rsystem}$ is more or less proportional to the fraction of scenarios ($g$-$w$ combinations) under which the system with that particular policy (a $C$-$y$ combination) results in a sustainable





outcome ($R_{system} > 0$). A similar concept has been used in the robust decision making literature (e.g., Groves and Lempert, 2007; Bryant and Lempert, 2010).

The surfaces and contours of the robustness of (or rather, inverse of) resilience, $\sigma_{Rsystem}$, associated with different policies over the policy space are shown in Figs. (3c and d), respectively. The $\sigma_{Rsystem}$ "landscape" is more

irregular, with two peaks (low robustness) and one trough (high robustness). The region with high robustness appears to be in the same general areas as the region with high resilience.

Figure 3: the mean, $\mu_{R-system}$, and the standard deviation, $\sigma_{R-system}$, of $R_{system}$ over entire decision space: (a) Surface of the resilience metric, $\mu_{R-system}$; (b) Contours of $\mu_{R-system}$; (c) Surface of the robustness, $\sigma_{R-system}$; (d) Contours of $\sigma_{R-system}$

We explore the interplay between $\mu_{Rsystem}$ and $\sigma_{Rsystem}$ in Fig. 4. Figure 4 shows that there are no perfect policies in the sense that no policies yield both maximum resilience and maximum robustness. Recall that the robustness indicates how sensitive $R_{system}$ itself is to external disturbances. The best policies are those along the Pareto frontier in the resilience-robustness space: among this set of Pareto-optimal policies, an increase in resilience is necessarily accompanied by a decrease in robustness, clearly illustrating the tradeoff between robustness and resilience. The

Pareto-optimal policies appear to be in two groups: those with high resilience and those with low resilience. Fig. 5 illustrates where the Pareto-optimal policies—with relatively high resilience—are located in the policy space.

Figure 4: Resilience-robustness tradeoff. Each point represents, $\mu_{R-system}$ and $\sigma_{R-system}$ of the coupled system with a given policy. Pareto-optimal policies are in two groups: those with relatively high resilience (red dots) and those with relatively low resilience (green dots).

Figure 5: Pareto optimal policies with high $\mu_{R-system}$ in the policy space ($C$-$y$ plane), superimposed with $\mu_{R-system}$ contours (a) and $\sigma_{R-system}$ contours (b).

## 4. Discussion and conclusions

In this paper, we exploit the simplicity of the stylized model to quantitatively define resilience and its robustness of a coupled infrastructure system. The resilience metric developed here is a measure of how far the CIS is from the

boundaries beyond which it will collapse. The model affords us with expressions of these boundaries, which clearly show how social and biophysical factors interplay to define these boundaries. With a concrete definition of resilience, resilience itself can be considered as the "of what" in the "robustness of what to what" notion. In particular, we use the standard deviation of the quantitatively defined resilience metric as the metric of robustness (low standard deviation means high robustness). Consequently, this enables us to rigorously investigate the interplay between the

two important, but not always well-defined, system properties. A key finding is the fundamental tradeoff between resilience and robustness: there are no perfect policies in governing a CIS, only Pareto-optimal ones.

Importantly, we hope for this work to stimulate further advances in rigorous studies of CISs, a few of which we briefly discuss here. More dimensions can be considered in defining Pareto-optimality. Figure 5 may give an impression that the set of Pareto-optimal policies is confined to a small region in the policy space, which would imply that PIPs



do not have that many choices—even in a simple CIS like the one studied. But that would be a wrong impression. In addition to resilience and robustness (as defined here), a policy maker or a social planner may be interested in other types of robustness with different "of what" and "to what" components. She may also be concerned about other system properties, e.g., productivity, user participation, etc. As more dimensions are considered, the set of Pareto-optimal policies grow. In the same spirit as that of the work done here, these other dimensions should be defined rigorously.

This work also lends itself to more rigorous studies of "adaptive governance." In the present study, the governance structure, represented by a policy (a combination of $C$ and $y$), is fixed. A natural next step is to explore if a policy is allowed to change, how one may improve the resilience and robustness of a CIS and/or alter the nature of their tradeoff. For example, if $C$ and $y$ are to be functions of other factors, e.g., resource availability and outside incentives, what functional forms should they take to improve the system's resilience and robustness?

In keeping with the theme of "social dynamics and planetary boundaries in Earth system modelling," the results shed light on how social and biophysical factors may interplay to define "boundaries" of a sustainable coupled system. While the modelled system here is admittedly simple, our methodology and results constitutes a step toward quantitatively and meaningfully combining social and biophysical factors into indicators of boundaries of more complex systems. Just as in this work, once those boundaries are clearly defined, calculation and discussion of resilience and robustness can become concrete and go beyond rhetoric and buzzwords.

**Author contribution:** MH and RM designed the study. MH and RM developed the analysis code and performed the simulations. MH, RM, and JMA wrote the manuscript.

**Competing interests:** The authors declare that they have no conflict of interest.

**Acknowledgements:** MH acknowledges the support from the University of Florida and its Department of Agricultural and Biological Engineering. RM and JMA acknowledge the support from National Science Foundation Grant GEO-1115054.

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

APPENDIX

**Basic model.** Here we brief describe the basic model presented by Muneepeerakul and Anderies (2017). The model

shows dynamic behaviour of three principal variables, namely, the state of the public infrastructure, $I_{HM}$, resource level, $R$, and the fraction of time user makes use of infrastructure, $U$, through Eq's (1, 4 and 5). In this context, $I_{HM}$ depends on PIPs in term of maintenance cost and has a positive relationship with the capacity of users to create resource flows. Equation (1) illustrates the dynamics of $I_{HM}$ as follows:

$$\frac{dI_{HM}}{dt} = M(\dots) - \delta H(I_{HM})$$

(A1)

where, $\delta$ is the infrastructure's depreciation rate and $H(I_{HM})$ states functional relationship of public infrastructure and productivity of each resource user. According to Muneepeerakul and Anderies, (2017) many shared infrastructures can be modeled by threshold functions. Given that $H(I_{HM})$ shows threshold behavior, they used a piecewise linear function to capture such behavior through Eq. (2).

$$H(I_{HM}) = \begin{cases} 0, & I_{HM} < I_0 \\ h\dfrac{I_{HM} - I_0}{I_m - I_0}, & I_0 \le I_{HM} \le I_m \\ h, & I_{HM} \ge I_m \end{cases}$$

(A2)

where, $h$ represents maximum amount of harvest by each user under no restriction and $I_0$ and $I_m$ are lower bound and upper bound thresholds of $I_{HM}$ respectively. Also, $M(\dots)$ is maintenance function (Eq. 3) and depends on social structure of the system.

$$M(\dots) = \mu_2 y C p N R U H(I_{HM})$$

(A3)



In Eq. (3), given the number of users $N$, $NRUH(I_{HM})$ is the total harvest from the natural infrastructure. As shown through Fig. 1, the resource users sell total harvest at price $p$ to generate revenue. Subsequently, they assign a proportion $C$ of revenue to PIP's for their contribution. Meanwhile, The PIP's spend proportion $y$ of $C$ on maintaining public infrastructure through the maintenance function $M(\dots)$. Also, $\mu_2$ is maintenance effectiveness of PIP's investment.

The second variable is resource level, $R$. They assumed the dynamics of resource to be:

$$\frac{dR}{dt} = G(R) - NURH(I_{HM})$$

(A4)

Natural infrastructure is assumed to invoke the conservation law comprising of regenerating capacity $(G(R) = g - dR)$ and total unit of harvest, $NURH(I_{HM})$. The definition presented for $G$ is the simplest model for natural infrastructure where $g$ and $d$ are the natural replenishment and the loss rates, respectively.

The strategic behavior of the resource users (RU's) is captured by employing replicator equation. Indeed, Replicator dynamics provide modeler with simple, realistic social mechanism where agents follow and replicate better-off strategies. The two possible strategies considered for RU's are staying inside system with the associated payoff of $\pi_U = (1 - C)pRH(I_{HM})$ or leaving system with the payoff of $\pi_w = w$. According to replicator equation:

$$\frac{dU}{dt} = rU(1 - U)(\pi_U - w)$$

(A5)

Replicator equation discuss the fraction of time that RU's assign to working inside system given $C$ and $y$. Like RU's, there is also two alternatives for PIP's, working inside system or working for another CIS which leads to system failure. Meanwhile, $C$ and $y$ characterize the strategy or policy of PIPs. The PIPs will participate in this coupled system only when $\pi_p = (1 - y)pCUNRH(I_{HM}) \geq \pi_w$. In other words, the PIPs maintain in the system when they are better-off than working outside. This condition is termed the PIP Participation Constraint (PPC).

Based on the system of three differential equations (Eqs. 1, 4 and 5), the sustainable equilibria, i.e., long-term system outcomes that satisfy the stability condition and PIP Participation Constraint (PPC), can be expressed as follows:

$$i_{HM}^* = \frac{yCR^*U^*N}{g}H(I_{HM}^*); \quad R^* = \frac{g}{d}\left(1 - \frac{i_{HM}^*}{yC}\right); \quad U^* = \frac{(1 - C)}{yC}\emptyset_1 i_{HM}^*$$

(A6)

where $i_{HM}^* \coloneqq \frac{I_{HM}^*\delta}{\mu_2 pg}[-]$ ([$-$] indicates dimensionless).



FIGURES

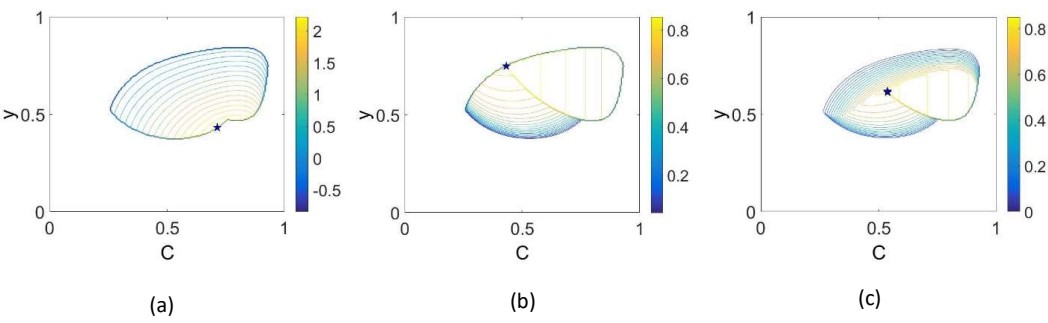

(a)      (b)      (c)

**Figure 1: Resilience metrics for a specific scenario (a $g$-$w$ combination) inside the sustainable region in the policy space (i.e., $C$-$y$ plane): (a) $R_{PPC}$ contours; (b) $R_{Stability}$ contours; and (c) $R_{system}$ contours.**

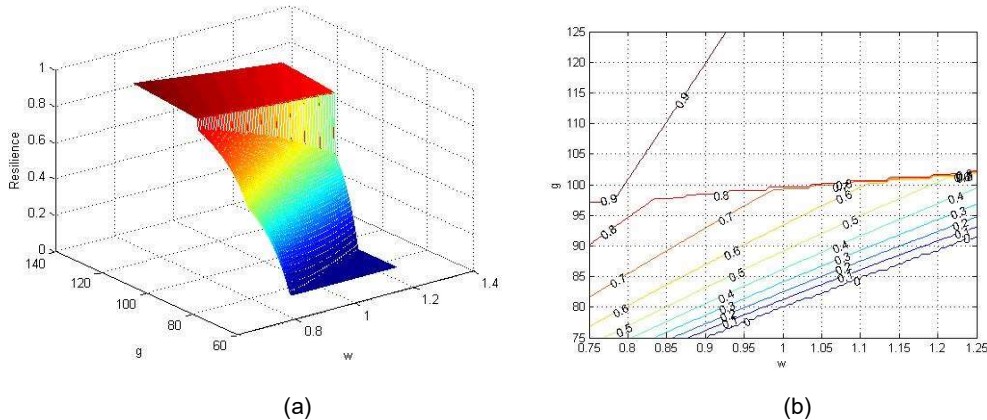

(a)                  (b)

**Figure 2: Variation of $R_{system}$ of a CIS with a fixed policy $(C, y)$ over 10,000 scenarios associated with a disturbance regime characterized by $\{g \in [75, 125], w \in [0.75, 1.25]\}$: (a) $R_{system}$ surface and (b) $R_{system}$ contours. The values of $R_{system}$ are used to calculate the mean, $\mu_{Rsystem}$, and the standard deviation, $\sigma_{Rsystem}$, of the variation of $R_{system}$. In this particular case, the resilience does not change much when $g$ is greater than about 100, but becomes more sensitive to both $g$ and $w$ when $g$ is lower than 100.**

15

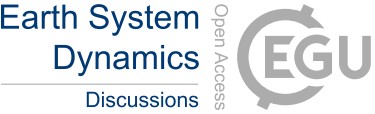

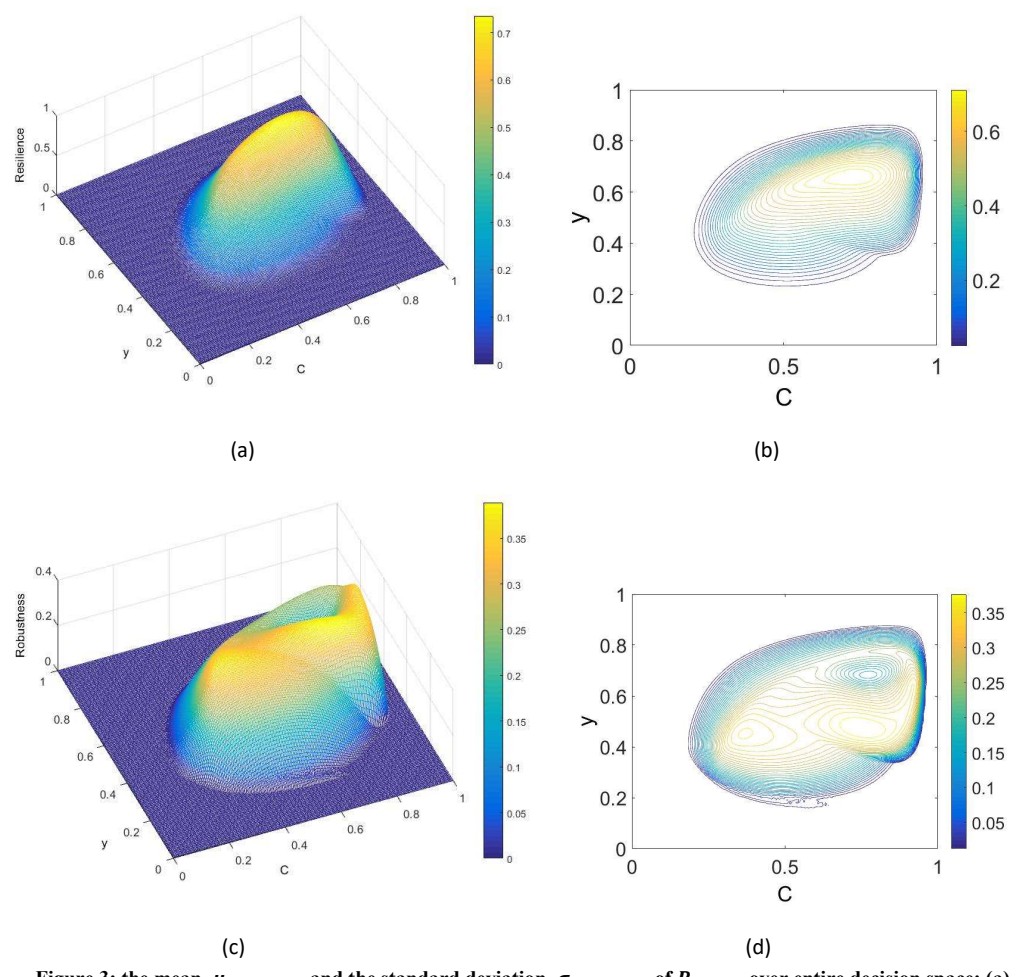

Figure 3: the mean, $\mu_{R-system}$, and the standard deviation, $\sigma_{R-system}$, of $R_{system}$ over entire decision space: (a) Surface of the resilience metric, $\mu_{R-system}$; (b) Contours of $\mu_{R-system}$; (c) Surface of the robustness, $\sigma_{R-system}$; (d) Contours of $\sigma_{R-system}$



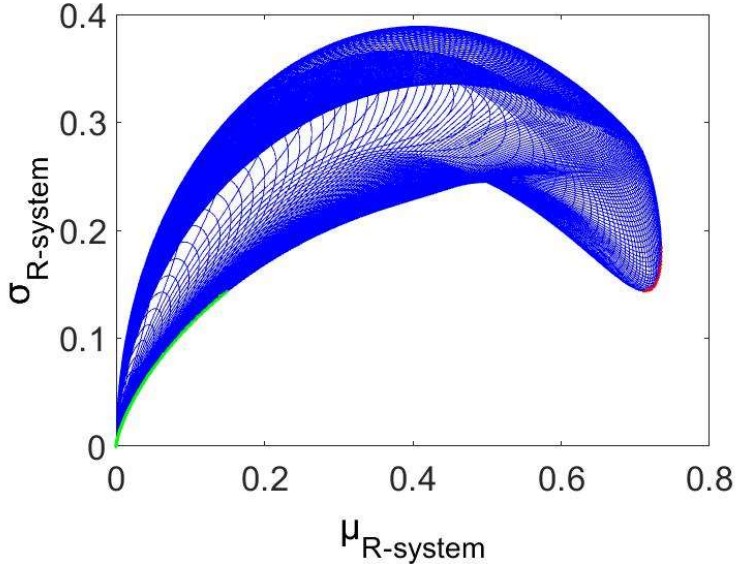

**Figure 4: Resilience-robustness tradeoff. Each point represents, $\mu_{R-system}$ and $\sigma_{R-system}$ of the coupled system with a given policy. Pareto-optimal policies are in two groups: those with relatively high resilience (red dots) and those with relatively low resilience (green dots).**

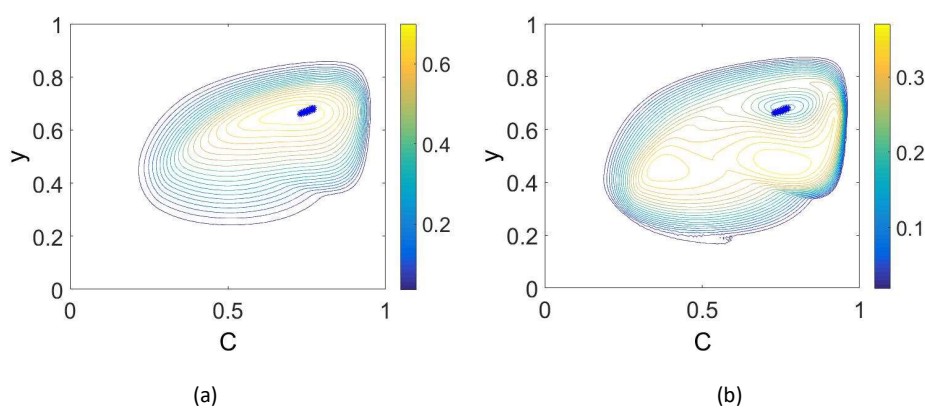

(a)                                                                 (b)

**Figure 5: Pareto optimal policies with high $\mu_{R-system}$ in the policy space ($C$-$y$ plane), superimposed with $\mu_{R-system}$ contours (a) and $\sigma_{R-system}$ contours (b).**

