# Peer review of "Linking Resilience and Robustness and Uncovering their Tradeoffs in Coupled Infrastructure Systems"

_Earth System Dynamics, 2017_

## Short Comment (SC1) · 15 Jan 2018

Dear reviewers and interested readers:

As you read the manuscript, please be aware of the following changes we plan to implement: a revised definition of the robustness metric and an addition of the author who suggested it.

Soon after the manuscript was submitted, I had a long and productive discussion with Chitsomanus P. Muneepeerakul about the manuscript. After much debate, we are convinced that using high variance of resilience to mean lower robustness could be misleading and, importantly, the variance should not be used as a dimension in determining Pareto-optimal policies. Her critique came from her research in the area of

environmental finance, where the use of variance to measure risk has been seen as problematic. She proposed that we use 'conditional mean' or what is referred to in the finance world as "conditional value at risk" as a robustness metric. For example, we may consider only resilience metric values lower than a certain threshold percentile and calculate the average of these "bad-tail values." The key objection here is that variance gives the same weight to "good deviations" (resilience greater than the mean) and "bad deviations" (resilience lower than the mean). Accordingly, she argued that, for example, contribution to a high variance arising from a heavy tail in the good direction should not be translated to less robustness. This issue has been a problem in the financial world, and more and more analyses have switched to considering other measures of risk, such as the conditional value at risk, in evaluating their portfolios of investment.

Following that logic, here we propose to use a "below-mean mean" as a new robustness metric, i.e., the mean of all resilience values associated with the "bad deviations." This new definition of the robustness metric has several desirable features. First, it can now be appropriately thought of as a robustness metric in the sense that the higher the value, the more robust the system (unlike the variance for which low variance means high robustness). Second, by using the mean as the threshold value for bad deviations, we remove some arbitrariness associated with prescribing a certain quantile (e.g., 5th or 10th quantile) in calculating the bad-tail mean. Third, it still carries some information about the sensitivity of the resilience metric to outside factors–the information that variance conveys; that is, the higher the "below-mean mean" (i.e., the bad deviations from the mean are small), the less sensitive the resilience metric.

We have implemented the new metric and it did perform better. In particular, using the new robustness metric yields a similar set of Pareto-optimal policies–without the not-so-desirable low-resilience, low-variance policies (which are technically part of the Pareto frontier when using variance as one of the dimensions to determine Pareto optimality) that we had to somewhat arbitrarily ignore in the original submission (Fig. 4).

Indeed, such arbitrary omission exemplifies the problem of using variance to measure robustness and to determine Pareto optimality. I am now of the opinion that the proposed metric is a better metric of robustness, not only for this particular model, but also for robustness of other indicator functions. This change will significantly strengthen our manuscript.

Your time and attention to these changes are appreciated.

Sincerely,

Rachata Muneepeerakul

On behalf of the authors

---

## Referee Comment (RC1) · Anonymous Referee #1 · 22 Mar 2018

This is an original research article that proposes two novel metrics of robustness and resilience. The authors identify a continual challenge with robustness and resilience: both these properties may require contextualisation. That is, how is robust and/or resilient X to Y. Here, the authors evaluate the relationship between their two novel measures of robustness and resilience. Specifically they explore trade-offs. So not just how does X or Y change as a consequence of changing inputs or assumptions. This has the potential for some significant impact.

The authors in the original submission argue that robustness is the more straightforward property to measure. Since submission the authors have effectively submitted a correction in that they have identified a limitation - or at least undesirable property - of their definition of robustness. Consequently I am not able to evaluate the quantitative Printer-friendly version

elements of the manuscript. Instead, I consider some general then specific points.

**General comments**

The work is based on a previously published model. To aid the reader in understanding the model I would suggest reproducing Figure 1 from Muneepeerakul, R. and Anderies, J.M., 2017. Strategic behaviors and governance challenges in social-ecological systems. Earth's Future.

P7 L4 "She may also be concerned about other system properties, e.g., productivity, user participation, etc. As more dimensions are considered, the set of Pareto- optimal policies grow. In the same spirit as that of the work done here, these other dimensions should be defined rigorously."

This is not a criticism of the manuscript, more of a general observation: defining and including other properties and so adding other dimensions introduces the potential of adding very different measures to the framework. The problem of evaluating "apples to pears". But beyond that there will be unavoidable normative inputs. As the authors note "she may also be concerned about other systems properties". How much which will she concerned? What respective weightings would be given to such concerns? Given freedom to weight different factors, there are potentially a very large number of Pareto-optimal solutions. The authors propose a way to transparently link potentially different properties within a single (or at least well defined) CIS. Explicitly identifying trade-offs is a potentially valuable approach. Rather than address this at the end of the paper Discussion and conclusions, perhaps it could be worked into the paper's main set of motivations?

P7L7 "In the present study, the governance structure, represented by a policy (a combination of ðİŘű and ðİŚę), is fixed. A natural next step is to explore if a policy is allowed to change, how one may improve the resilience and robustness of a CIS and/or alter the nature of their tradeoff."
This is another potential valuable contribution. If one possible impact of this work is to inform policy, then there needs to be an ability to evaluate changes and adaptations of policy in the light of new knowledge. How robustness and resilience changes over time as a consequence of changing policy is a very important research question. I think the authors could argue that in the absence of transparent measure and metrics, attempts to explore adaptive policy may be importantly limited.

**Specific comments**

"The concepts of "resilience" and "robustness" have grown considerably in popularity as desirable properties for a 20 wide range of systems. Terms like "resilient communities" and "robust cities" have been used more frequently in public discourse."

This requires some evidential support. I also note that some of the key literature is potentially missing. For example, for a discussion on resilience on social-ecological systems, I would expect a reference to material such as Folke, C., 2006. Resilience: The emergence of a perspective for social–ecological systems analyses. Global environmental change, 16(3), pp.253-267. This would help address the first point as this article alone has been cited over 4300 times. Also, more recent work may be required. Holling (1973) is used to initially define resilience. The literature review stops at around 2006/7. Much work has been done since then. Another Folke paper for example: Folke, C., Carpenter, S., Walker, B., Scheffer, M., Chapin, T. and Rockström, J., 2010. Resilience thinking: integrating resilience, adaptability and transformability. Ecology and society, 15(4).

P1 I29 "Robustness may very well be a desirable property of a system, but it seems to come with a price."

The article is importantly about trade-offs. Perhaps also useful to simply point out that this may be an example of no free lunch? Wolpert, D.H. and Macready, W.G., 1997. No free lunch theorems for optimization. IEEE transactions on evolutionary computation, 1(1), pp.67-82.
P2 footnote - I would propose this text is integrated into the main text. Clearly and consistently define all terms.

P2 L16 SES is not defined.

P3 L7 typo: "(cite report of engineers)"

P4 I14 "Routh-Horowitz" this is either a typo "Routh-Hurwitz" or the authors are referring to some other work. In either event, a citation is required here.

Caption text for Figure1 and Figure 2 require explanation of star symbol.

Figure 4 red dots are very hard to see. Also bear in mind that RGB colour blindness may mean it is not possible to discern the difference between the red and blue.

---

## Referee Comment (RC2) · Anonymous Referee #2 · 26 Mar 2018

The present manuscript focuses on the relevant concepts of 'robustness' and 'resilience'. The authors revisit the concepts and aim for their proper quantification as well as study the connections between them. In doing so, they present an analytical framework based on a stylized dynamical model [proposed by Muneepeerakul and Anderies (2017)] that realizes a conceptual framework for socio-ecological systems [coupled infrastructure systems (CISs)] to formulate the setting for their proposals and analyses presented in the manuscript.

The authors set forth the boundaries of conditions for the sustainable operation of the aforementioned system. The system may collapse by crossing one of the two boundaries relating to: - a scenario where there is an over-arching requirement of investment for maintaining the infrastructure, such that there is not enough revenue

from the system. In this case, the system may be abandoned for an alternative one, - a scenario where the non-trivial equilibrium state of the system is unstable representing unsustainable maintenance of public infrastructure.

The authors define the measures of robustness and resilience relating to the above boundaries and study the connections between these measures. This reveals certain trade-offs between robustness and resilience of the system, which they relate to choices of certain policies implemented by social agents (e.g., taxation and investment in public infrastructure), internal stresses and external disturbances of the dynamical model considered.

The scope and results of the manuscript are potentially interesting and motivating. The authors hint at the potential implications of their results in 'understanding the interplay between social dynamics and planetary boundaries', whereby crossing the latter may drive the system to undesirable regimes. In this regard, the trade-off between robustness and resilience of a system can be particularly useful for assisting decision makers in governing and/or managing CISs. However, the presentation of the results in the manuscript needs improvement. In my opinion, the article is still premature for publication, but may definitely be considered after an appropriate revision. Also, the authors themselves have realized that variance (as used in the present version of the manuscript) is not an apt measure of robustness since it weighs above mean and below mean values of resilience equally. Thus, further discussion may be postponed until the revised results are presented in the subsequent version of the manuscript.

Specific comments:

1. The authors may consider rephrasing the title to: 'Robustness and Resilience - Quantification, Connections and Trade-offs' to have it more precise and well-rounded.

2. Page 2, line number 13: At this point, the authors may add references to more latest approaches in this direction (in addition to the suggestions by Anonymous Referee #1), such as that taken by - Mitra et al. (2015), An integrative quantifier of multistability in

complex systems based on ecological resilience, Scientific Reports, 5, 16196.

3. Page 3, line number 19: Why are these two conditions referred to as 'boundaries'? Please motivate or rephrase the terminology.

4. Page 3, line number 25: Why only the distance to the boundaries is considered as an effective measure here? What about the overall dynamics of the system within the phase space, or what about engineering resilience?

5. Page 5, line number 17: What is the reasoning behind choosing the ranges of uncertainties in 'g' and 'w' that the authors have used in the manuscript, namely, [75, 125] and [0.75, 1.75] respectively? Please state, if there is any physical reasoning underlying the above choice in order to motivate the reader about the same.

Technical comments:

1. Page 2, line number 16: Please abbreviate SESs prior to their use at this point and thereafter.

2. Page 3, line number 8: Please insert an appropriate reference here ;).

3. Page 3, line number 12: 'of' which 'the' key variables...

4. Page 4, line number 14: Is the Routh-Hurwitz stability criterion being referred to here? If yes, the necessary correction should be made along with the inclusion of a reference in this regard.

5. Page 9, line number 14: The word 'brief' should read as 'briefly'.

6. Page 9, line number 16: 'Eq's (1, 4 and 5)' should read as 'Eqs. (A1, A4 and A5)'.

7. Page 9, line number 22: The comma after 'Muneepeerakul and Anderies' is unnecessary.

8. Page 9, line number 25: The equation number '(A2)' should be indented properly within the same line.

9. Page 10, line number 2: Which Fig. 1 do the authors refer to here - of the manuscript under consideration or that of Muneepeerakul and Anderies (2017)?

10. Page 10, line number 3: 'The' should not be capitalized.

11. Page 10, line number 13: 'Replicator' should not be capitalized.

12. Page 10, line number 24: 'Eqs. 1, 4 and 5' should read as 'Eqs. (A1, A4 and A5)'.

13. This is a general comment concerning all the figures presented in the manuscript. The resolution of all the figures has to be substantially improved in the revised version of the manuscript. The 'x' and 'y' labels of the figures should be made larger in certain figures as well as the sizes of the texts associated with the colour bars in certain figures where they appear.

14. Figure 1: What do the stars represent?

15. Figure 2: What do the colours in Figure 2(a) represent - the standard deviation ($\mu_{system}$)? Otherwise, how are the standard deviations in this figure represented?

---

## Author Comment (AC1) · 23 Apr 2018

Dear ESD Editors: Attached is the revised version of our manuscript entitled "Linking Resilience and Robustness and Uncovering their Trade-offs in Coupled Infrastructure Systems" (formerly entitled "How robust is your system resilience?") in which we have incorporated and addressed the comments and concerns raised by the two reviewers. We wish to thank the editor and the two referees for the constructive comments which improve the manuscript and provide useful ideas for future work. Our detailed responses to the referees' comments are below, in a blue font. We wish to highlight some major changes here. As explained in our online discussion post, we have added Chitsomanus P. Muneepeerakul as an author due to her contribution on the new definition of robustness. We, naturally, replaced all the results based the old definition of

robustness with those based on the new definition. One of Referee #2's suggestions prompted a discussion among the authors, which resulted in the title being changed to "Linking Resilience and Robustness and Uncovering their Trade-offs in Coupled In-frastructure Systems." We followed the suggestion by Referee #1 and included a figure from Muneepeerakul and Anderies (2017) in the Appendix. This inclusion makes the present work more self-contained. Throughout the manuscript, we have revised the language to be more careful and precise in using such terms as "disturbance," "uncertainty," "scenario" and "setting." Our point-by-point responses to other comments are listed below. With these revisions, we believe the manuscript is much improved and now ready for publication. We again thank the editor and reviewers for their constructive and useful comments. Please do not hesitate to let me know if you have questions.

Sincerely,

Mehran Homayounfar On behalf of all authors -

Response to Referee #1 This is an original research article that proposes two novel metrics of robustness and resilience. The authors identify a continual challenge with robustness and resilience: both these properties may require contextualization. That is, how is robust and/or resilient X to Y. Here, the authors evaluate the relationship between their two novel measures of robustness and resilience. Specifically, they explore trade-offs. So not just how does X or Y change as a consequence of changing inputs or assumptions. This has the potential for some significant impact. The authors in the original submission argue that robustness is the more straightforward property to measure. Since submission the authors have effectively submitted a correction in that they have identified a limitation - or at least undesirable property – of their definition of robustness. Consequently, I am not able to evaluate the quantitative elements of the manuscript. Instead, I consider some general then specific points. General comments The work is based on a previously published model. To aid the reader in understanding the model I would suggest reproducing Figure 1 from Muneepeerakul, R. and Anderies, J.M., 2017. Strategic behaviors

ESDD
and governance challenges in social-ecological systems. Earth's Future. Response 1-1. Thank you for this suggestion. We have now included Figure 2 from Muneepeerakul and Anderies (2017), which is more explicitly connected to the equations, in Appendix. P7 L4 "She may also be concerned about other system properties, e.g., productivity, user participation, etc. As more dimensions are considered, the set of Pareto- optimal policies grow. In the same spirit as that of the work done here, these other dimensions should be defined rigorously."

This is not a criticism of the manuscript, more of a general observation: defining and including other properties and so adding other dimensions introduces the potential of adding very different measures to the framework. The problem of evaluating "apples to pears". But beyond that there will be unavoidable normative inputs. As the authors note "she may also be concerned about other systems properties". How much which will she concerned? What respective weightings would be given to such concerns? Given freedom to weight different factors, there are potentially a very large number of Pareto-optimal solutions. The authors propose a way to transparently link potentially different properties within a single (or at least well defined) CIS. Explicitly identifying trade-offs is a potentially valuable approach. Rather than address this at the end of the paper Discussion and conclusions, perhaps it could be worked into the paper's main set of motivations?

Response 1-2. We share the reviewer's perspective that indeed considering resilience and robustness together is like comparing apples to pears. This is why our analysis stops at the Pareto frontier—without giving relative weights to either property. In other words, we simply present a fundamental tradeoff between two very different properties, and it is up to the policy maker who "gives weights"—either explicitly by giving numeric weights to these properties or implicitly/subconsciously by the act of selecting a policy. If the weights are given, this problem will reduce to an optimization problem, and the rich nature of tradeoffs would be lost or at least overshadowed. As the reviewer pointed out, it is the tradeoffs that we wish to highlight. In fact, we already included the need to Interactive comment

identify such tradeoffs as one of our motivations for this study (P2, L6).

P7L7 "In the present study, the governance structure, represented by a policy (a combination of C and y), is fixed. A natural next step is to explore if a policy is allowed to change, how one may improve the resilience and robustness of a CIS and/or alter the nature of their tradeoff."

This is another potential valuable contribution. If one possible impact of this work is to inform policy, then there needs to be an ability to evaluate changes and adaptations of policy in the light of new knowledge. How robustness and resilience changes over time as a consequence of changing policy is a very important research question. I think the authors could argue that in the absence of transparent measure and metrics, attempts to explore adaptive policy may be importantly limited.

Response 1-3. Thank you very much for this comment. We have inserted a sentence along the suggested line into the Discussion (P8, L20).

**Specific comments**

"The concepts of "resilience" and "robustness" have grown considerably in popularity as desirable properties for a wide range of systems. Terms like "resilient communities" and "robust cities" have been used more frequently in public discourse." This requires some evidential support. I also note that some of the key literature is potentially missing. For example, for a discussion on resilience on social-ecological systems, I would expect a reference to material such as Folke, C., 2006. Resilience: The emergence of a perspective for social–ecological systems analyses. Global environmental change, 16(3), pp.253-267. This would help address the first point as this article alone has been cited over 4300 times. Also, more recent work may be required. Holling (1973) is used to initially define resilience. The literature review stops at around 2006/7. Much work has been done since then. Another Folke paper for example: Folke, C., Carpenter, S., Walker, B., Scheffer, M., Chapin, T. and Rockström, J., 2010. Resilience thinking: integrating resilience, adaptability and transformability. Ecology and society, 15(4).
Response 1b-1. The suggested references and several additional ones are now cited. Thank you for your suggestions.

P1 I29 "Robustness may very well be a desirable property of a system, but it seems to come with a price." The article is importantly about trade-offs. Perhaps also useful to simply point out that this may be an example of no free lunch? Wolpert, D.H. and Macready, W.G., 1997. No free lunch theorems for optimization. IEEE transactions on evolutionary computation, 1(1), pp.67-82.

Response 1b-2. Thank you for this suggestion. The suggested reference is now cited.

P2 footnote - I would propose this text is integrated into the main text. Clearly and consistently define all terms.

Response 1b-3. The text has been moved into the main text (P2, L22).

P2 L16 SES is not defined.

Response 1b-4. It is now defined. Thanks for catching this.

P3 L7 typo: "(cite report of engineers)"

Response 1b-5. Properly cited now. P3, L14.

P4 I14 "Routh-Horowitz" this is either a typo "Routh-Hurwitz" or the authors are referring to some other work. In either event, a citation is required here.

Response 1b-6. The spelling is corrected, and relevant references cited. P4-L30.

Caption text for Figure1 and Figure 2 require explanation of star symbol.

Response 1b-7. The figure captions now include "the black star in panels (a), (b), and (c) indicate the policy with the highest R\_PPC, R\_Stability, and R\_system, respectively."

Figure 4 red dots are very hard to see. Also bear in mind that RGB colour blindness may mean it is not possible to discern the difference between the red and blue.

**ESDD**
Response 1b-8. This figure does not need to be in color. We have replaced the red dots with bigger, black dots, and the blue dots with smaller, gray dots.

Response to Referee #2 The present manuscript focuses on the relevant concepts of 'robustness' and 'resilience'. The authors revisit the concepts and aim for their proper quantification as well as study the connections between them. In doing so, they present an analytical framework based on a stylized dynamical model [proposed by Muneepeerakul and Anderies (2017)] that realizes a conceptual framework for socio-ecological systems [coupled infrastructure systems (CISs)] to formulate the setting for their proposals and analyses presented in the manuscript.

The authors set forth the boundaries of conditions for the sustainable operation of the aforementioned system. The system may collapse by crossing one of the two boundaries relating to: - a scenario where there is an over-arching requirement of investment for maintaining the infrastructure, such that there is not enough revenue from the system. In this case, the system may be abandoned for an alternative one, - a scenario where the non-trivial equilibrium state of the system is unstable representing unsustainable maintenance of public infrastructure. The authors define the measures of robustness and resilience relating to the above boundaries and study the connections between these measures. This reveals certain trade-offs between robustness and resilience of the system, which they relate to choices of certain policies implemented by social agents (e.g., taxation and investment in public infrastructure), internal stresses and external disturbances of the dynamical model considered.

The scope and results of the manuscript are potentially interesting and motivating. The authors hint at the potential implications of their results in 'understanding the interplay between social dynamics and planetary boundaries', whereby crossing the latter may drive the system to undesirable regimes. In this regard, the trade-off between robustness and resilience of a system can be particularly useful for assisting decision makers in governing and/or managing CISs. However, the presentation of the results in the manuscript needs improvement. In my opinion, the article is still premature for
publication, but may definitely be considered after an appropriate revision. Also, the authors themselves have realized that variance (as used in the present version of the manuscript) is not an apt measure of robustness since it weighs above mean and below mean values of resilience equally. Thus, further discussion may be postponed until the revised results are presented in the subsequent version of the manuscript. Specific comments:

The authors may consider rephrasing the title to: 'Robustness and Resilience - Quantification, Connections and Trade-offs' to have it more precise and well-rounded.

Response 2-1. We have taken the Referee's suggestion into account and modified our title to "Linking Resilience and Robustness and Uncovering their Trade-offs in Coupled Infrastructure Systems."

Page 2, line number 13: At this point, the authors may add references to more latest approaches in this direction (in addition to the suggestions by Anonymous Referee #1), such as that taken by - Mitra et al. (2015), An integrative quantifier of multistability in complex systems based on ecological resilience, Scientific Reports, 5, 16196.

Response 2-2. The suggested reference and additional ones are now cited. (P2, L18)

Page 3, line number 19: Why are these two conditions referred to as 'boundaries'? Please motivate or rephrase the terminology.

Response 2-3. The two conditions dissect the PIP decision space (i.e., the C-y plane) into different regions, namely the regions in which the coupled system can be sustained or collapses. Graphically, the two conditions are represented by curves that form the boundaries between the sustainable and collapse regions in the decision space.

Page 3, line number 25: Why only the distance to the boundaries is considered as an effective measure here? What about the overall dynamics of the system within the phase space, or what about engineering resilience?

Response 2-4. The Referee is right that there are different facets of resilience. Yes,

**ESDD**
the engineering resilience, or the recovery-based resilience, is another way that people think of resilience. In this study, we focus on the regime shift-based resilience (traditionally called ecological resilience) and its tradeoff with robustness, which we believe is novel and requires expositional clarity. Presenting several metrics of resilience, let alone studying their tradeoffs with perhaps different kinds of robustness, we fear, would make the arguments too confusing to follow and dilute the key messages we wish to convey. In any event, we have added text to clarify the scope of our work, i.e., the focus on the regime shift-based resilience. P3-L34.

Page 5, line number 17: What is the reasoning behind choosing the ranges of uncertainties in 'g' and 'w' that the authors have used in the manuscript, namely, [75, 125] and [0.75, 1.75] respectively? Please state, if there is any physical reasoning underlying the above choice in order to motivate the reader about the same.

Response 2-5. This is a theoretical study that highlights the interplay and tradeoff between resilience and robustness. The model is not based on any particular system, but rather a generic one. The ranges of g and w were simply chosen so that the ranges of results exhibit the pattern of the resilience-robustness tradeoff well. There was no physical reasoning behind the parameter ranges beyond that.

Technical comments:

Page 2, line number 16: Please abbreviate SESs prior to their use at this point and thereafter.

Response 2b-1. Done.

Page 3, line number 8: Please insert an appropriate reference here ;).

Response 2b-2. Properly cited now. P3-L14.

Page 3, line number 12: 'of' which 'the' key variables...

Response 2b-3. That clause has been removed.
Page 4, line number 14: Is the Routh-Hurwitz stability criterion being referred to here? If yes, the necessary correction should be made along with the inclusion of a reference in this regard.

Response 2b-4. The spelling is corrected, and relevant references cited. P4-L30.

Page 9, line number 14: The word 'brief' should read as 'briefly'.

Response 2b-5. Accordingly changed.

Page 9, line number 16: 'Eq's (1, 4 and 5)' should read as 'Eqs. (A1, A4 and A5)'.

Response 2b-6. Accordingly changed.

Page 9, line number 22: The comma after 'Muneepeerakul and Anderies' is unnecessary.

Response 2b-7. Removed. P11-L24.

Page 9, line number 25: The equation number '(A2)' should be indented properly within the same line.

Response 2b-8. Accordingly changed.

Page 10, line number 2: Which Fig. 1 do the authors refer to here - of the manuscript under consideration or that of Muneepeerakul and Anderies (2017)?

Response 2b-9. A figure from Muneepeerakul and Anderies (2017) has been included in the Appendix as Figure A1 and is now referred to properly.

Page 10, line number 3: 'The' should not be capitalized.

Response 2b-10. Accordingly changed.

Page 10, line number 13: 'Replicator' should not be capitalized.

Response 2b-11. Accordingly changed.
Page 10, line number 24: 'Eqs. 1, 4 and 5' should read as 'Eqs. (A1, A4 and A5)'.

Response 2b-12. Accordingly changed.

This is a general comment concerning all the figures presented in the manuscript. The resolution of all the figures has to be substantially improved in the revised version of the manuscript. The 'x' and 'y' labels of the figures should be made larger in certain figures as well as the sizes of the texts associated with the colour bars in certain figures where they appear.

Response 2b-13. We have replaced some of the figures with those of better quality.

Figure 1: What do the stars represent?

Response 2b-14. The figure captions now include "the black star in panels (a), (b), and (c) indicate the policy with the highest R\_PPC, R\_Stability, and R\_system, respectively."

Figure 2: What do the colours in Figure 2(a) represent - the standard deviation (nmu\_system)? Otherwise, how are the standard deviations in this figure represented?

Response 2b-15. The colors represent essentially the same information as the surface and contours: red means high resilience and dark blue means low resilience. There are no standard deviations in this figure: each point on the surface and contours represents the resilience metric of the fixed policy under a given social-ecological setting (formerly termed "scenario"—a combination of fixed g and w—which is determined by evaluating the deterministic dynamical system model.

Please also note the supplement to this comment: https://www.earth-syst-dynam-discuss.net/esd-2017-124/esd-2017-124-AC1supplement.zip Interactive comment

---

## Author Response (AR2)

Dear ESD Editors:

Attached is the revised version of our manuscript entitled "Linking Resilience and Robustness and Uncovering their Trade-offs in Coupled Infrastructure Systems" in which we have incorporated and addressed the second round of comments and concerns raised by the two reviewers. We wish to thank the editor and the two referees for the constructive comments which improve the manuscript and provide useful ideas for future work.

Our point-by-point responses to other comments are listed below. With these revisions, we believe the manuscript is much improved and now ready for publication. We again thank the editor and reviewers for their constructive and useful comments. Please do not hesitate to let me know if you have questions.

Sincerely,

*Mehran Homayounfar*

On behalf of all authors

**Response to Referee #2**

1- Page 1, line number 26: I suggest moving the (web page-based) reference to the bibliography at the end of the main text.

**Response 2-1.** Thank you for this suggestion. We have moved the reference to the bibliography at the end of the main text as suggested (P1:L26 and P11:L40).

2- Consider a scenario where the distribution of R_{system} is highly skewed towards the right. This would result in a high value of the mean, giving the impression of a high value of the resilience than it practically should. Also, a larger part of the distribution would be available for computing \mu_{R < \mu} which could subsequently yield a misleading value of the robustness. A related problem could arise when the distribution of R_{system} is highly skewed to the left. The measures proposed in the revised manuscript and the analysis revolving around them seem to be all affected by the aforementioned issue. Please clarify your stand on this problem and if necessary, add details in the next iteration of the manuscript for dealing with it.

**Response 2-2.** Thank you for this comment. The Reviewer raised a good point—one that applies generally to finding good measures of central tendency and risk for a random variable with a highly skewed distribution, not just our metric of resilience. This issue would be more critical if we were to use or interpret the *absolute* values of $R_{system}$ directly. However, in this study, we use $\mu_{Rsystem}$ and $\mu_{R<\mu}$ to systematically compare across systems with different policies.

An example would help here. Consider a system with policy $\{C_1, y_1\}$, which yields, under 10,000 settings, 10,000 values of $R_{system}$ that are more or less normally distributed. Consider another system with policy $\{C_2, y_2\}$. Suppose now that for 5,000 settings, this second system yields the same lower-half values of $R_{system}$ as those of the first system. However, for the remaining 5,000 settings, the second system yields $R_{system}$ values that are three times those of the first system. The distribution of the second system's $R_{system}$ is thus skewed to the right, leading to higher $\mu_{Rsystem}$ and $\mu_{R<\mu}$. We would then say that the second system is superior to the first both in terms of resilience and robustness, and it would be a logical conclusion, as under no settings is the second system is less resilient than the first. (On the other hand, if the variance is used, one would say the second system is superior to the first in one dimension, i.e., that of robustness, which is somewhat illogical since, again, under no settings is the second system is less resilient than the first.)

In sum, we believe that for their usage in our study—to compare and highlight trade-offs among systems with different policies—$\mu_{Rsystem}$ and $\mu_{R<\mu}$ are useful and appropriate metrics.

3- Page 7, line number 10: What do the two local maxima of \mu_{R < \mu} actually reflect, i.e., are there greater implications of these maxima in the context of the stylized dynamical model considered in the manuscript?
**Response 2-3.** They simply reflect the nonlinear interplay between the model parameters and model structure and may affect the nature of the trade-off between robustness and resilience reported in Figures 4 and 5. We have added a remark on this issue on P7:L17.

4- Page 7, line number 33: The authors mention - 'In particular, we use the standard deviation of the quantitatively defined resilience metric as the metric of robustness (low standard deviation means high robustness).' I suspect that the authors have forgotten to revise this statement from the previous version of the manuscript. If yes, please appropriately revise it.

**Response 2-4.** Thank you for catching this typo! We have revised the manuscript accordingly (P8:L8).

**Response to Referee #3**

Review of the revised version of "Linking Resilience and Robustness and Uncovering their Trade-offs in Coupled Infrastructure Systems" (formerly entitled "How robust is your system resilience?")". In general, the manuscript is rather clear and concise, and the work is relevant and interesting enough to merit publication. However, I do have some reservations that I would like to point out before acceptance.

First, although I wholeheartedly agree that we should put more effort in specifying and quantifying resilience and related properties, the study focuses on engineering resilience only. This is a result of the choice to use a dynamical system model in which the behavior of agents is included only in a very indirect and inflexible way. The assumption in this CIS model is that agents operate with perfect knowledge, foresight, and decision-making regarding investments in and yields from infrastructure (replicator dynamics). I have no problem with that, as long as it is recognized as such. I think there is a third type of resilience that is not recognized as such in this manuscript, and that is resilience resulting from adaptation. Agents in a CIS may have the capability of changing their behaviour in response to policy changes, environmental shocks, technical change, etc., which may also change the structure (and not only parameter values) of the CIS. It remains to be seen in how far the used methodology is applicable when agent behaviour is included explicitly, in particular when agents change their behaviour and decision-making after policy changes. That said, I do not feel the authors have to cover all aspects in one paper, but in my view paragraph 16-21 of the discussion on page 8 should be rewritten as it currently oversteps the application range of the resilience metric used in this manuscript. Exploring policy effects is very relevant, but should include a model construct that allows for agent behavioural change or at least an explicit description of the current assumptions on agent behaviour regarding policy.

**Response 3-1.** Thank you for the comment. Regarding the definition of resilience, the current study focuses in fact on "ecological resilience," as indicated in the text P3, L31-33: "*Here, it is worth noting that, while it is possible to examine recovery-based resilience (or the so-called "engineering resilience"), this paper focuses on regime shift-based resilience (traditionally called "ecological resilience") and its robustness.*" In any event, the Reviewer pointed out important issues regarding adaptation, which is related to resilience. We have added an additional paragraph below to discuss this issue in the 'discussion and conclusions' section (starting on P8:L30).

"Additionally, agents in a CIS may have the capacity to change their behaviour in response to changes in policy, environmental conditions, technological changes, and the like. In this study, strategic behaviour and decision-making process are assumed unchanged in the analysis. Adaptation in strategic behaviour of agents will subsequently alter the nature of resilience, its robustness, and their trade-offs. Capturing such effects of adaptation requires

structural changes to the model, e.g., in terms of specification of payoffs or even the formulation of the dynamical equations. With such adaptive social agents, how should one devise adaptive governance to enhance resilience and robustness of a CIS? Addressing such a question is a theoretically intriguing future research direction with great practical implications."

Second, I have some questions about the model and the model analysis. Is there an overview of the model variables, parameters, and their nominal values, ranges, dimensions, units and meaning? This would help in getting a direct overview of the model, and I would welcome such an addition to the appendix on the model description.

**Response 3-2.** We provide brief description of the model in the Appendix. A more detailed description is available in an earlier paper (Muneepeerakul & Anderies, Earth's Future 2017); we did not want—nor did we think the Editor would think it is appropriate—to repeat too many details from another already published paper.

Of somewhat bigger concern is the model analysis. The authors use Routh-Hurwitz criteria to determine the resilience regarding stability. The model however includes nonlinear terms, in particular an Allee model-like term $rU(1 - U)(\pi_U - w)$, and seems to resemble food web models from the field of theoretical ecology. Are there no bifurcations in the model other than tangent bifurcations that mark the boundaries of resilience? I find it hard to assess, as there are no parameter values reported, and I cannot analyse the model in-depth myself. I think this is not the case here, otherwise the landscape pictures would probably look differently, but I would think this is something that could be clarified given the potential relevance for the resilience assessment of the model.

**Response 3-3.** The Reviewer is right in that there may exist other interesting bifurcations in the model. However, here we focus on those related to the non-trivial sustainable long-term outcome. Analyzing other bifurcations related to other equilibria may be mathematically interesting, but it could make the manuscript less accessible and dilute the key message about the resilience-robustness trade-off of different policies aiming at keeping the system in the basin of attraction of the non-trivial sustainable long-term outcome. We have added a remark on this issue (P4:L32).

The parameter values are the same as those used in the previous work (Muneepeerakul & Anderies, Earth's Future 2017) and now included in the revised manuscript (P13:L18). Thank you for pointing this out to us.

Minor comments:

Page 2, end of the top paragraph does not read very well. It refers to the desirability of resilience, but resilience has not been introduced yet. Maybe switch this piece of text with the first lines of the following paragraph in which resilience is explained?

**Response 3-4.** The ending part of the paragraph is a way to set up the research question of the study. It aims at invoking the reader's curiosity about the potential trade-off between resilience and robustness—regardless of what the reader's own definitions of these terms might be. Our strategy is to get the reader interested first and then follow by a more concrete treatment of these terms. We feel that this flow of narrative works well and would like to keep it.

Page 6, between lines 10 and 15, it could be explained already there to what values is referred to with "those values associated with 'bad deviations'". It becomes clear later on in the third paragraph of the page, but it does not read very well.

**Response 3-5.** Here, we agree with the Reviewer that referring to "bad deviations" here may cause unnecessary confusion and not read well. We have removed the phrase in the revised manuscript (P6:L13).

Figures 4 and 5: I think you could add a line explaining the black dots.

**Response 3-6.** Thank you for this comment. The figures' captions have been updated accordingly (P7:L29-32 and P15 and 16).

Figure 5 caption typo: mu_R-syste, without the m.

**Response 3-7.** We in fact could not find the typo.

Why is there variation in the description of UNRH(I_HM)? Eq. (A3): NRUH; Eq. (A4): NURH; Fig. A1: UNRH.

**Response 3-8.** Thank you for this comment. The equations have been updated accordingly.

What is the meaning of the slashed O in Eq. (A6)?

**Response 3-9.** This was the Greek letter phi ($\phi$), whose definition has been now included (P13-L18). Thank you for catching this oversight.

**Linking Resilience and Robustness and Uncovering their Trade-offs in Coupled Infrastructure Systems**

Mehran Homayounfar[1], Rachata Muneepeerakul[1], John M. Anderies[2], and Chitsomanus P. Muneepeerakul[3]

[1]Department of Agricultural and Biological Engineering, University of Florida, Gainesville, Florida, USA
[2]School of Sustainability and School of Human Evolution and Social Change, Arizona State University, Tempe, Arizona, USA
[3]Independent researcher, Gainesville, Florida, USA

*Correspondence to*: Mehran Homayounfar (homayounfar@ufl.edu) and Rachata Muneepeerakul (rmuneepe@ufl.edu)

**Abstract.** Robustness and resilience are concepts in systems thinking that have grown in importance and popularity. For many complex social-ecological systems, however, robustness and resilience are difficult to quantify and the connections and trade-offs between them difficult to study. Most studies have either focused on qualitative approaches to discuss their connections or considered only one of them under particular classes of disturbances. In this study, we present an analytical framework to address the linkage between robustness and resilience more systematically. Our analysis is based on a stylized dynamical model that operationalizes a widely used conceptual framework for social-ecological systems. The model enables us to rigorously delineate the boundaries of conditions under which the coupled system can be sustained in a long run, define robustness and resilience related to these boundaries, and consequently investigate their connections. The results reveal the trade-offs between robustness and resilience. They also show how the nature of such trade-offs varies with the choices of certain policies (e.g., taxation and investment in public infrastructure), internal stresses and uncertainty in social-ecological settings.

**1. Introduction**

The concepts of "resilience" and "robustness" have grown considerably in popularity as desirable properties for a wide range of systems. Terms like "resilient communities" and "robust cities" have been used more frequently in public discourse (e.g., Chang and Shinozuka, 2004; Longstaff et al., 2010; Chang et al., 2014). The UK's Water Act 2014 even included "primary duty to secure resilience" as one of the general duties of its Water Services Regulation Authority (Water Act, 2014). Growing with that popularity is some confusion and potential misuse of the terms "robustness" and "resilience" due to imprecision, vagueness, and multiplicity of their definitions. Such lack of consistency and rigor hinders advances in our understanding of the interplay between these two important system properties.

Relatively speaking, robustness has been defined more consistently and rigorously—as it can be linked to a more familiar concept of sensitivity. For example, according to Carlson and Doyle (2002), robustness in engineering systems refers to the maintenance of system performance either when subjected to external disturbances or internal uncertain parameters. In other words, in robust systems, performance is less sensitive to disturbances or uncertainty.

Robustness may very well be a desirable property of a system, but it seems to come with a price. Recent research shows that tuning a system to be robust against certain disturbance regimes almost always reduces system performance and likely increases its vulnerability to other disturbance regimes (Ostrom et al., 2007; Anderies et al., 2007; Bode, 1945; Csete and Doyle, 2002; Wolpert and Macready, 1997). Now, if resilience is also a desirable property of the same system, does it also come at the expense of performance and robustness? Put it another way, is there a trade-off among performance, robustness, and resilience? Such a trade-off, if exists, is a crucial consideration for governing and/or managing social-ecological systems (SESs).

But resilience, as alluded to above, is trickier to define. According to Holling (1973), resilience refers to the amount of change or disruption required to shift the maintenance of a system along different sets of mutually reinforcing processes and structures. In other words, resilience can be thought of as how far the system is from certain thresholds or boundaries beyond which the system will undergo a regime shift or a quantitative change in system structure or identity. Holling (1996) categorized resilience into two types, engineering resilience, which refers to the ability of a system to return to steady state following a perturbation, and ecological resilience, which refers to the capacity of system to remain in a particular stability domain in the face of perturbations. The latter category is used by many researchers to discuss resilience of SESs, or more generally, *coupled infrastructure systems* (CISs) (Carpenter et al., 2001; Folke, S et al., 2002; Anderies et al., 2006; Folke, 2006; Folke et al., 2010; Biggs et al., 2012; Barrett and Constas, 2014; Redman, 2014; Walker et al.,2002; Gunderson et al., 1995; Berkes and Folke, 1998; Carpenter et al., 1999a, 1999b; Scheffer et al., 2000; Berkes et al., 2003; Walker et al., 2004; Carpenter and Brock, 2004; Janssen et al., 2004; Folke et al., 2002; Anderies et al., 2006; Folke et al., 2016; Cote and Nightingale, 2012; Mitra et al., 2015; Cumming and Peterson, 2017). The term coupled infrastructure systems (CISs) has been introduced to generalize the notions of coupled natural-human systems (CNHSs) and social-ecological systems (SESs); in this context, *infrastructure* is broadly defined to include human-made, social, and natural infrastructure (see, e.g., Anderies et al., 2016). The problem is that these CISs are complex and thus identifying thresholds and potential regime shifts associated with those thresholds is often difficult, if not impossible. In many cases, major aspects of resilience in CISs may not be directly observable and must be actualized indirectly via surrogate attributes (Carpenter et al., 2005; Kerner and Thomas, 2014). Recent significant advances have been made toward identifying early-warning signals that indicate whether a critical threshold is being approached for a wide class of systems (Scheffer et al., 2009 and 2012). Still, there are gaps in our understanding of how indicators of resilience and robustness will behave in more complex situations. This lack of a rigorous metric for resilience makes the investigation into their connections, interplay, and trade-offs with robustness and performance impossible.

But these knowledge gaps need to be filled if one wishes to make advances in understanding the interplay between social dynamics and planetary boundaries. Given the magnitude of impacts that human activities have on pushing Earth systems toward their planetary boundaries, we need clearer understanding of how social and biophysical factors come together to define the nature of these boundaries. This paper is a step in that direction. In particular, we will build on recent work that mathematically operationalizes the Robustness of SES framework (Anderies et al., 2004) into a formal stylized dynamical model (Muneepeerakul and Anderies, 2017). We will exploit the relative simplicity

of the model to rigorously define robustness and resilience of the coupled system. The modelled system will be subject to fluctuations in external drivers, which will affect the well-defined robustness and resilience, thereby enabling us to investigate the interplay and trade-offs between these important properties, as well as how the nature of the interplay and trade-offs are affected by policies implemented by social agents.

5  **2. Methods**

Here we analyse a mathematical model developed by Muneepeerakul and Anderies (2017) by subjecting the coupled system to uncertainty in ecological and social factors. The model captures the essential features of a system in which a group of agents shares infrastructure to produce valued flows. Such a system is the archetype of most, if not all of human sociality: groups produce infrastructure that they cannot produce individually (security, defence, irrigation

10  canals, roads, markets, financial systems, coordination mechanisms, etc.) that significantly increases productivity. The challenge is maintaining this shared infrastructure (e.g. decaying infrastructure is a major problem in the US at the time of writing (ASCER CIA Advisory Council, 2013). The model allows for mathematical definitions of the boundaries of policy domain that result in a sustainable system in which both human-made and natural infrastructure can be maintained over the long run. Based on these boundaries and uncertainty in the exogenous factors, we define

15  metrics of resilience and robustness associated with each policy choice and investigate the trade-off between them. The basic model presented by Muneepeerakul and Anderies (2017) is described in the Appendix.

Here a policy is defined as a combination of taxation level $C$ and the proportion of tax revenue invested in infrastructure maintenance $y$ that the public infrastructure providers (PIPs) decide to implement in the system. The infrastructure (e.g. canals) enable resource users (RUs) to produce valued goods from a natural resource. The two

20  fluctuating exogenous factors are the replenishment rate of the natural resource $g$ and the wage $w$ that resource users (RUs) would earn from working outside the system—a combination of $g$ and $w$ defines a "social-ecological setting" or simply "setting." There are two boundaries that, once crossed, will cause the system will collapse. The first boundary is called PIP participation constraint (PPC): when the PIPs must invest too much in maintaining the public infrastructure (exceeding the opportunity cost of $w_P$) and/or cannot retain enough revenue for themselves, they will

25  abandon the system for another. The second boundary is the stability condition of the non-trivial equilibrium point (i.e., the "society" in which both PIPs (e.g. the state) and RUs (e.g. citizens) participate in the system and public infrastructure is sufficiently maintained in a long run). Together, these two boundaries delineate a set of policies ($C$-$y$ combinations) that correspond to sustainable outcomes. The resilience metric to be developed below can be thought of as a metric of how far the system is from these boundaries. As the two exogenous factors defining settings, namely

30  $g$ and $w$, fluctuate, the two boundaries and thus the resilience metric, too, fluctuate with them. How sensitive the resilience metric is to these fluctuating settings is used to define robustness. Here, it is worth noting that, while it is possible to examine recovery-based resilience (or the so-called "engineering resilience"), this paper focuses on regime shift-based resilience (traditionally called "ecological resilience") and its robustness. Quantification of and trade-offs between resilience and robustness is a novel concept that requires expositional clarity. Presenting several metrics of

35  resilience, let alone studying their trade-offs with potentially different metrics of robustness, may confuse the matter

and dilute the key messages we attempt to convey. As such, in what follows, we will focus on developing a metric for regime shift-based resilience. With the scope of analysis clarified and suitably bounded, we will now define resilience and robustness more formally.

**2.1. Resilience metric**

5    Direct measurement of above-mentioned resilience, as a specified form of resilience (Walker et al., 2004), in SES's is difficult because boundaries and thresholds that separate domains of dynamics for SES's are difficult to identify (Carpenter et al., 2005; Scheffer et al., 2009 and 2012). In this stylized model, however, such boundaries can be clearly identified by the stability condition (SC) and the PPC. Here, we are interested in the resilience of system's ability to provide sufficient livelihoods for the PIPs and resource users. The basin of attraction for system resilience is defined

10    by those system states (i.e. infrastructure state) in which this is possible, and these system states are directly mapped to the SC and PPC. We will thus define resilience metrics based on the SC and PPC boundaries. Here our goal is to develop resilience metrics that can be meaningfully compared to one another. As such, we identify some desired properties that guide the definitions of these resilience metrics. First, they should be zero at their respective boundaries. Second, positive values indicate greater resilience of the system in a desirable state. These first two properties align

15    with how resilience has been measured, i.e., the distance from the boundary of a basin of attraction (e.g., Anderies et al., 2002; S. R. Carpenter et al., 1999). Third, to facilitate the consideration of relative risks associated with different types of regime shifts that the system may be facing, the metrics should be comparable in magnitude. These properties guide us toward the following definitions of the resilience metrics.

We define the resilience of the system against abandonment by PIPs as follows:

$$R_{PPC} = (\pi_{PIP}/w_P) - 1, (1)$$

where $\pi_P$ is the net revenue that PIPs collect and $w_P$ is the opportunity cost that they will earn if they choose to work with another system. Positive values of $R_{PPC}$ indicate that the system is resilient against being abandoned by PIPs, while negative values indicate that the system will eventually collapse due to the PIPs' abandonment. It is important to note that $\pi_P$ results from the coupled dynamics of the CIS; this means that $R_{PPC}$ has already integrated the dynamics

25    of infrastructure, resource, and resource users (Eqs. A1, A4 and A5), making it a metric of the system, not of an individual component.

Numerical analysis of the model indicates that the equilibrium becomes unstable when the following Routh-Hurwitz condition (e.g., May, 2001; Kot, 2001) is violated:

$$D - T(J_{1,1}J_{2,2} + J_{2,1}J_{1,2} + J_{2,3}J_{3,2} + J_{1,3}J_{3,1}) > 0, (2)$$

30    where, $D$, $T$, and $J's$ are determinant, trace, and entries, respectively, of the Jacobian matrix of the dynamical system (Eqs. A1, A4, and A5) evaluated at the nontrivial equilibrium point (Eq. A6)—when such an equilibrium point exists. Here, it is worth noting that we focus on the equilibrium point related to the non-trivial sustainable long-term outcome. Analyzing other bifurcations related to other equilibria may be mathematically interesting, but it could make the study less accessible and dilute its key message about the resilience-robustness trade-off of different policies aiming at

35    keeping the system in the basin of attraction of the non-trivial sustainable long-term outcome.

Following the guideline provided by the three desirable properties above, we rearrange terms in Eq. (2) and define the resilience of the system against instability (increased probability of collapse of infrastructure) as follows:

$$R_{stability} = \frac{D}{\left|T\left(J_{1,1}J_{2,2} + J_{2,1}J_{1,2} + J_{2,3}J_{3,2} + J_{1,3}J_{3,1}\right)\right|} - 1, (3)$$

This formulation is parallel to that of the first resilience metric (Eq. 1); it possesses the three properties: $R_{stability}$ of zero indicates the boundary between stability and instability; positive $R_{stability}$ means the system at the equilibrium point is stable; and the magnitudes of $R_{stability}$ are comparable to those of $R_{PPC}$ (see Fig. 1). Note that $R_{stability}$, too, is determined from the coupled dynamics of the CIS; this means that it has integrated the dynamics of infrastructure, resource, and resource users (Eqs. A1, A4 and A5).

This allows us to meaningfully define the overall system resilience as the minimum between the two resilience metrics, namely:

$$R_{system} = \begin{cases} Min[R_{PPC}, R_{Stability}], \wedge R_{PPC}, R_{stability} \geq 0 \\ 0, \wedge \; otherwise \end{cases}, (4)$$

Equation (4) implies that $R_{system}$ is positive only when the nontrivial equilibrium point (Eq. A6) exists and is stable; otherwise, the system is considered not resilient and denoted by $R_{system} = 0$. $R_{system}$ thus represents the tension between the PIPs being too greedy (high $C$, low $y$) whereby they get close to the stability boundary and "not greedy enough," i.e., low $C$ and high $y$ whereby they get close to the PPC, given a particular choice for $w_P$. Note that the values of $w$ and $w_P$ represent the socio-economic embedding of the CIS. Therefore, the biophysical structure of the CIS along with the socio-economic context in which it is embedded co-determine the maximum resilience that can be achieved. Given that the nontrivial equilibrium point exists and is stable, if the system is at a greater risk of being abandoned by the PIPs (and eventually collapsing), $R_{system} = R_{PPC}$; if the system is at a greater risk of becoming unstable (and eventually collapsing), $R_{system} = R_{stability}$. Figure 1 illustrates the relationships between $R_{PPC}$, $R_{stability}$, and $R_{system}$.

Figure 1: Resilience metrics for a specific setting (a $g$-$w$ combination) inside the sustainable region in the policy space (i.e., $C$-$y$ plane): (a) $R_{PPC}$ contours; (b) $R_{Stability}$ contours; and (c) $R_{system}$ contours. The black star in panels (a), (b), and (c) indicate the policy with the highest $R_{PPC}$, $R_{Stability}$, and $R_{system}$, respectively.

**2.2. Linking Robustness and resilience**

As discussed earlier, robustness can be thought of as the opposite of sensitivity. A commonly used measure of sensitivity is variance. Thus, variance of a given function under specific disturbance or uncertainty regimes may be used to indicate robustness of that function against those disturbance or uncertainty regimes (robustness of what to what). In this case, the system function of interest is the system resilience $R_{system}$. By choosing $R_{system}$ as our function, we can usefully link these concepts. If we use a ball and cup metaphor for resilience, the robustness of

resilience refers to the degree at which the geometry of the cup changes as a result of external disturbances and/or parameter changes. However, as we will argue below, relating high variance of $R_{system}$ to low robustness may be misleading and should not be used in evaluating a given policy. By definition, the variance treats "good deviations" and "bad deviations" from the mean equally. For functions with preferred values, such as resilience or profit, values greater than the mean and those lower should not be treated in the same way. Specifically, contribution to a high variance from a heavy tail in the good direction should not be translated to less robustness. This problem also arises in assessing financial risk: what makes an asset risky is the values on the "bad tail" of the distribution (i.e., low or negative profits). This has motivated more and more analyses to switch to considering other measures of risk, such as the conditional value at risk, in evaluating their portfolios of investment (Rockafellar and Uryasev, 1999; Krokhmal et al., 2001; Sarykalin et al., 2008; Zymler et al., 2013). Intuitively, this means that the shape of the "cup" can be asymmetric and we need to take this into account.

Following this logic, we propose to use a "below-mean mean" as a new robustness metric: the mean of all resilience values lower than the mean, i.e., those values associated with the "bad deviations.". This new definition of the robustness metric has several desirable features. First, it can now be appropriately thought of as a robustness metric in the sense that the higher the value, the more robust the system (unlike the variance for which low variance means high robustness). Second, by using the mean as the threshold value for bad deviations, we remove some arbitrariness associated with prescribing a certain quantile (e.g., 5th or 10th quantile) in calculating the conditional value at risk. Third, it still carries some information about the sensitivity of the resilience metric to outside factors—the information that variance conveys; that is, the higher the "below-mean mean" (i.e., the bad deviations from the mean are small and the below-mean mean is close to the mean), the less sensitive—and thus more robust—the resilience metric.

In this study, we subject the modelled system to uncertainty in one natural factor and one social factor, namely, the natural replenishment rate of the resource $g$, and the payoff that a RU earns from working outside the system $w$. Thus, we are computing how the resilience of the system to shocks/variation in state variables changes as the parameters $g$ and $w$ change (i.e., we are uncertain about the underlying social-ecological setting of the system). In particular, we assume that $g$ is uniformly distributed over the range [75, 125] and $w$ is uniformly distributed over the range [0.75, 1.75]. A social-ecological setting, or setting, is defined as a combination of $g$ and $w$. For a given policy (a $C - y$ combination), we calculate $R_{system}$ for 10,000 settings (i.e., 10,000 $g - w$ combinations) (see Fig. 2). Then, from these 10,000 values of the resilience metric $R_{system}$, we calculate the mean, $\mu_{R-system} = E[R_{system}]$, and use it as the **resilience metric** of the coupled system with a given policy, and the below-mean mean, $\mu_{R<\mu} = E[R_{system} \vee R_{system} < \mu_{R-system}]$, as the **metric for robustness of resilience.** This metric measures the robustness of the capacity of the system to cope with variation in state variables $I_{HM}, R,$ and $U$ to fundamental uncertainty about the underlying setting of the system.

Figure 2: Variation of $R_{system}$ of a CIS with a fixed policy $(C, y)$ over 10,000 settings associated with uncertainty characterized by $\{g \in [75,125], w \in [0.75,1.25]\}$: (a) $R_{system}$ surface and (b) $R_{system}$ contours. The values of $R_{system}$ are used to calculate the mean, $\mu_{Rsystem}$, and the below-mean mean, $\mu_{R<\mu}$. In this particular case, the resilience does not change much when $g$ is greater than about 100, but becomes more sensitive to both $g$ and $w$ when $g$ is lower than 100.

**3. Results**

The surfaces and contours of the system resilience metric, $\mu_{Rsystem}$, and associated with different policies $(C - y)$ over the policy space are shown in Figs. (3a and b), respectively. The policies with sustainable outcomes are located in the middle of the policy space, with $\mu_{Rsystem}$ peaking in the center and declining as policies become more extreme in either direction. Our analysis also shows that $\mu_{Rsystem}$ is more or less proportional to the fraction of settings ($g$-$w$ combinations) under which the system with that particular policy (a $C$-$y$ combination) results in a sustainable outcome ($R_{system} > 0$). A similar concept has been used in the robust decision making literature (e.g., Groves and Lempert, 2007; Bryant and Lempert, 2009).

The surfaces and contours of the robustness of, $\mu_{R<\mu}$, associated with different policies over the policy space are shown in Figs. (3c and d), respectively. The $\mu_{R<\mu}$ "landscape" is more irregular, having two local maxima with one being more dominant than the other. The region with high robustness appears to be in the same general areas as the region with high resilience. These features reflect the nonlinear interplay between the model parameters and model structure and may affect the nature of the trade-off between robustness and resilience reported in Figures 4 and 5.

Figure 3: the mean, $\mu_{R-system}$, and the below-mean meanof $R_{system}$, $\mu_{R<\mu}$, over entire decision space: (a) Surface of the resilience metric, $\mu_{R-system}$; (b) Contours of $\mu_{R-system}$; (c) Surface of the robustness, the below-mean mean ($\mu_{R<\mu}$); (d) Contours of the robustness, the below-mean mean ($\mu_{R<\mu}$)

We explore the interplay between $\mu_{Rsystem}$ and $\mu_{R<\mu}$ in Fig. 4. Figure 4 shows that there are no perfect policies in the sense that no policies yield both maximum resilience and maximum robustness. Recall that the robustness indicates how sensitive $R_{system}$ itself is to uncertainty in the underlying setting of the system (e.g., $g$ and $w$). The best policies are those along the Pareto frontier in the resilience-robustness space: among this set of Pareto-optimal policies, an increase in resilience is necessarily accompanied by a decrease in robustness, clearly illustrating the trade-off between robustness and resilience. Fig. 5 illustrates where the Pareto-optimal policies are located in the policy space.

Figure 4: Resilience-robustness trade-off. Each point represents, $\mu_{R-system}$ and $\mu_{R<\mu}$ of the coupled system with a given policy. The trade-off is only apparent at the Pareto-optimal frontier (the blaeck dots represent a set of Pareto-optimal policies).

Figure 5: Pareto optimal policies, represented by (the blaeck dots,) with high in the policy space ($C$-$y$ plane), superimposed with resilience ($\mu_{R-system}$) contours (a) and robustness ($\mu_{R<\mu}$) contours (b).

**4. Discussion and conclusions**

In this paper, we exploit the simplicity of a stylized model to quantitatively link resilience and robustness by computing how the CIS's resilience to shocks in state variables changes with parameters. In this way, we compute the robustness of CIS resilience to uncertainty in the underlying CIS setting. The resilience metric developed here is a measure of how far the CIS is from the boundaries beyond which it will collapse. The model affords us with expressions of these boundaries, which clearly show how social and biophysical factors interplay to define these boundaries. With a concrete definition of resilience, resilience itself can be considered as the "of what" in the "robustness of what to what" notion. In particular, we use the  mean  of the quantitatively defined resilience metric as the metric of robustness . Consequently, this enables us to rigorously investigate the interplay between the two important, but not always well-defined, system properties. A key finding is the fundamental trade-off between resilience and robustness: there are no perfect policies in governing a CIS, only Pareto-optimal ones. Specifically, policies designed to maximize the resilience of a CIS to shocks on timescales at which the state variables play out may be very sensitive to being wrong about our understanding of the underlying dynamics of the CIS in question.

Importantly, we hope this work will stimulate further advances in rigorous studies of CISs that address such subtle, policy-relevant questions, a few of which we briefly discuss here. More dimensions can be considered in defining Pareto-optimality. Figure 5 may give an impression that the set of Pareto-optimal policies is confined to a small region in the policy space, which would imply that PIPs do not have that many choices—even in a simple CIS like the one studied. But that would be a wrong impression. In addition to resilience and robustness (as defined here), a policy maker or a social planner may be interested in other types of robustness with different "of what" and "to what" components. She may also be concerned about other system properties, e.g., productivity, user participation, etc. As more dimensions are considered, the set of Pareto-optimal policies grow. In the same spirit as that of the work done here, these other dimensions should be defined rigorously.

This work also lends itself to more rigorous studies of "adaptive governance." In the present study, the governance structure, represented by a policy (a combination of $C$ and $y$), is fixed. A natural next step is to explore if a policy is allowed to change, how one may improve the resilience and robustness of a CIS and/or alter the nature of their inherent trade-offs. For example, if $C$ and $y$ are to be functions of other factors, e.g., resource availability and outside incentives, what functional forms should they take to improve the system's resilience and robustness? Indeed, in the absence of transparent metrics, attempts to explore such adaptive policies are severely limited.

Additionally, agents in a CIS may have the capacity to change their behaviour in response to changes in policy, environmental conditions, technological changes, and the like. In this study, strategic behaviour and decision-making process are assumed unchanged in the analysis. Adaptation in strategic behaviour of agents will subsequently alter the nature of resilience, its robustness, and their trade-offs. Capturing such effects of adaptation requires structural changes to the model, e.g., in terms of specification of payoffs or even the formulation of the dynamical equations.

With such adaptive social agents, how should one devise adaptive governance to enhance resilience and robustness of a CIS? Addressing such a question is a theoretically intriguing future research direction with great practical implications.

In keeping with the theme of "social dynamics and planetary boundaries in Earth system modelling," our results shed light on how social and biophysical factors may interplay to define "boundaries" of a sustainable coupled infrastructure system. While the modelled system here is admittedly simple, our methodology and results constitute a step toward quantitatively and meaningfully combining social and biophysical factors into indicators of boundaries of more complex systems. Just as in this work, once those boundaries are clearly defined, calculation and discussion of resilience and robustness can become concrete.

**Author contribution:** MH, RM, JMA, and CPM designed the study. MH carried out the analysis. MH and RM prepared the manuscript with contributions from all co-authors.

**Competing interests:** The authors declare that they have no conflict of interest.

**Acknowledgements:** JMA and RM acknowledge the support from the grant NSF GEO-1115054. The authors also thank the editor and two anonymous referees for their constructive and useful comments.

**5. Reference:**

Anderies, John M., et al.: Grazing Management, Resilience, and the Dynamics of a Fire-Driven Rangeland System., Ecosystems, vol. 5, no. 1, pp. 23–44, doi:10.1007/s10021-001-0053-9, **2002**.

Anderies, J., Janssen, M., & Schlager, E.: Institutions and the performance of coupled infrastructure systems. International Journal of the Commons, 10(2), **2016**.

Anderies, J., et al.: A Framework to Analyze the Robustness of Social-Ecological Systems from an Institutional Perspective. Ecology and Society, http://www.ecologyandsociety.org/vol9/iss1/art18/inline.html, **2004**.

Anderies, J. M., et al.: Panaceas, Uncertainty, and the Robust Control Framework in Sustainability Science. Proceedings of the National Academy of Sciences, vol. 104, no. 39, pp. 15194–99, doi:10.1073/pnas.0702655104, **2007**.

Anderies, John M., Paul Ryan, et al.: Loss of Resilience, Crisis, and Institutional Change: Lessons from an Intensive Agricultural System in Southeastern Australia. Ecosystems, vol. 9, no. 6, pp. 865–78, doi:10.1007/s10021-006-0017-1, **2006**.

ASCE RCIA Advisory Council: Report card for America's Infrastrcutre, Tech. Rep., American Society of Civil Engineers, **2013**.

Barrett, Christopher B., and Mark A. Constas.: Toward a Theory of Resilience for International Development Applications. Proceedings of the National Academy of Sciences, vol. 111, no. 40, pp. 14625–30, doi:10.1073/pnas.1320880111, **2014**.

Berkes, Fikret, et al.: Navigating Social-Ecological Systems: Building Resilience for Complexity and change Building, p. 393, doi:10.1016/j.biocon.2004.01.010, **2003**.

Berkes, Fikret, and Carl Folke.: Linking Social and Ecological Systems for Resilience and Sustainability. Linking Social and Ecological Systems: Management Practices and Social Mechanisms for Building Resilience, vol. 1, pp. 13–20, **1998**.

Biggs, Reinette, et al.: Toward Principles for Enhancing the Resilience of Ecosystem Services. Annual Review of Environment and Resources, vol. 37, no. 1, Nov, pp. 421–48, doi:10.1146/annurev-environ-051211-123836,

**2012**.

Bode, Hendrik Wade.: Network Analysis and Feedback Amplifier Design. Bell Telephone Laboratories Series, p. 551, doi:10.1183/09031936.00138507, **1945**.

Bryant, Benjamin P., and Robert J. Lempert.: Thinking inside the Box: A Participatory, Computer-Assisted Approach to Scenario Discovery. Technological Forecasting and Social Change, vol. 77, no. 1, pp. 34–49, doi:10.1016/j.techfore.2009.08.002, **2010**.

Carlson, J. M., and J. Doyle.: Complexity and Robustness. Proceedings of the National Academy of Sciences, vol. 99, no. Supplement 1, pp. 2538–45, doi:10.1073/pnas.012582499, **2002**.

Carpenter, S., et al.: From Metaphor to Measurement: Resilience of What to What? Ecosystems, http://www.springerlink.com/index/lu5mxqkxbblkly62.pdf, **2001**.

Carpenter, S. R., et al.: Management of Eutrophication for Lakes Subject to Potentially Irreversible Change. Ecological Applications, vol. 9, no. 3, pp. 751–71, doi:10.2307/2641327, **1999**.

Carpenter, Stephen R., William Brock, et al.: Ecological and Social Dynamics in Simple Models of Ecosystem Management. *Conservation Ecology*, vol. 3, no. 2, pp. 1–31, doi:10.5751/ES-00122-030204, **1999**.

Carpenter, Stephen R., Frances Westley, et al.: Surrogates for Resilience of Social-Ecological Systems. *Ecosystems*, vol. 8, no. 8, pp. 941–44, doi:10.1007/s10021-005-0170-y, **2005**.

Carpenter, Stephen R., and William A. Brock.: Spatial Complexity, Resilience, and Policy Diversity: Fishing on Lake-Rich Landscapes. Ecology and Society, vol. 9, no. 1, **2004**.

Chang, Stephanie E., et al.: Toward Disaster-Resilient Cities: Characterizing Resilience of Infrastructure Systems with Expert Judgments. Risk Analysis, vol. 34, no. 3, pp. 416–34, doi:10.1111/risa.12133, **2014**.

Chang, Stephanie E., and Masanobu Shinozuka.: Measuring Improvements in the Disaster Resilience of Communities. Earthquake Spectra, vol. 20, no. 3, pp. 739–55, doi:10.1193/1.1775796, **2004**.

Cote, Muriel, and Andrea J. Nightingale.: Resilience Thinking Meets Social Theory. Progress in Human Geography, vol. 36, no. 4, pp. 475–89, doi:10.1177/0309132511425708, **2012**.

Csete, Marie E., and John C. Doyle.: Reverse Engineering of Biological Complexity. Science (New York, N.Y.), vol. 295, no. 5560, pp. 1664–69, doi:10.1126/science.1069981, **2002**.

Cumming, Graeme S., and Garry D. Peterson.: Unifying Research on Social–Ecological Resilience and Collapse. Trends in Ecology and Evolution, vol. 32, no. 9, Elsevier Ltd, pp. 695–713, doi:10.1016/j.tree.2017.06.014, **2017**.

Folke, Carl.: Resilience: The Emergence of a Perspective for Social–ecological Systems Analyses. Global Environmental Change, vol. 16, no. 3, Aug., pp. 253–67, doi:10.1016/j.gloenvcha.2006.04.002, **2006**.

Folke, Carl, Steve Carpenter, et al.: Resilience and Sustainable Development: Building Adaptive Capacity in a World of Transformations. AMBIO: A Journal of the Human Environment, vol. 31, no. 5, p. 437, doi:10.1639/0044-7447(2002)031[0437:RASDBA]2.0.CO;2, **2002**.

Folke, Carl, Stephen R. Carpenter, et al.: Resilience Thinking: Integrating Resilience, Adaptability and Transformability. Ecology and Society, vol. 15, no. 4, doi:10.5751/ES-03610-150420, **2010**.

Folke, Carl, Reinette Biggs, et al.: Social-Ecological Resilience and Biosphere-Based Sustainability Science. Ecology and Society, vol. 21, no. 3, doi:10.5751/ES-08748-210341, **2016**.

Groves, David G., and Robert J. Lempert.: A New Analytic Method for Finding Policy-Relevant Scenarios. Global Environmental Change, vol. 17, no. 1, pp. 73–85, doi:10.1016/j.gloenvcha.2006.11.006, **2007**.

Gunderson, Lance H., et al. Barriers and Bridges to the Renewal of Ecosystems and Institutions. Columbia University Press, **1995**.

Holling, C. S.: Resilience and Stability of Ecological Systems. Annual Review of Ecology and Systematics, vol.

4, no. 1, Nov, pp. 1–23, doi:10.1146/annurev.es.04.110173.000245, **1973**.

Holling, Crawford Stanley.: Engineering Resilience versus Ecological Resilience. Engineering Within Ecological Constraints, no. 1996, pp. 31–44, doi:10.17226/4919, **1996**.

Janssen, Marco A., et al.: Robust Strategies for Managing Rangelands with Multiple Stable Attractors. Journal of Environmental Economics and Management, vol. 47, no. 1, pp. 140–62, doi:10.1016/S0095-0696(03)00069-X, **2004**.

Kerner, David, and J. Thomas.: Resilience Attributes of Social-Ecological Systems: Framing Metrics for Management. Resources, vol. 3, no. 4, Multidisciplinary Digital Publishing Institute, pp. 672–702, doi:10.3390/resources3040672, **2014**.

Kot, M.: Elements of Mathematical Ecology. Cambridge University Press, **2001**.

Krokhmal, Pavlo, et al. Portfolio Optimization With Conditional Value-at-Risk Objective and Constraints. no. 352, pp. 1–30, **2001**.

Longstaff, Patricia H. H., et al.: Building Resilient Communities: A Preliminary Framework for Assessment." Homeland Security Affairs, vol. 6, no. 3, pp. 1–23, **2010**.

May, R. M.: Stability and Complexity in Model Ecosystems. Princeton university press, **2001**.

Mitra, Chiranjit, et al.: An Integrative Quantifier of Multistability in Complex Systems Based on Ecological Resilience. Scientific Reports, vol. 5, Nature Publishing Group, pp. 1–10, doi:10.1038/srep16196, **2015**.

Muneepeerakul, Rachata, and John M. Anderies. "Strategic Behaviors and Governance Challenges in Social-Ecological Systems." Earth's Future, vol. 5, no. 8, Wiley Periodicals, Inc., pp. 865–76, doi:10.1002/2017EF000562, **2017**.

Ostrom, E., et al.: Going beyond Panaceas. Proceedings of the National Academy of Sciences, vol. 104, no. 39, pp. 15176–78, doi:10.1073/pnas.0701886104, **2007**.

Redman, Charles L.: Should Sustainability and Resilience Be Combined or Remain Distinct Pursuits? Ecology and Society, vol. 19, no. 2, p. art37, doi:10.5751/ES-06390-190237, **2014**.

Rockafellar, R. Tyrrell, and Stanislav Uryasev. Optimization of Conditional Value-at-Risk, pp. 1–26, **1999**.

Sarykalin, Sergey, et al. Value-at-Risk vs. Conditional Value-at-Risk in Risk Management and Optimization, pp. 270–94, doi:10.1287/educ.1080.0052, **2008**.

Scheffer, Marten, Stephen R. Carpenter, et al.: Anticipating Critical Transitions. Science, vol. 338, no. 6105, pp. 344–48, doi:10.1126/science.1225244, **2012**.

Scheffer, Marten, Jordi Bascompte, et al.: Early-Warning Signals for Critical Transitions. Nature, vol. 461, no. 7260, Nature Publishing Group., pp. 53–59, doi:10.1038/nature08227, **2009**.

Scheffer, Marten, William Brock, et al.: Socioeconomic Mechanisms Preventing Optimum Use of Ecosystem Services: An Interdisciplinary Theoretical Analysis. Ecosystems, vol. 3, no. 5, pp. 451–71, doi:10.1007/s100210000040, **2000**.

Walker, B., et al.: Resilience Management in Social-Ecological Systems: A Working\nhypothesis for a Participatory Approach. Conservation Ecology [Online], vol. 6, no. 1, p. 14, doi:14, **2002**.

Walker, Brian, et al.: Resilience , Adaptability and Transformability in Social-Ecological Systems Resilience , Adaptability and Transformability in Social – Ecological Systems. Ecology and Society, vol. 9, no. 2, doi:10.5751/ES-00650-090205, **2004**.

Water act 2014, "Explanatory Notes have been produced to assist in the understanding of this Act and are available separately". CHAPTER 21, published by TSC (The Stationary Office) and available from: Online www.tsoshop.co.uk; (http://www.legislation.gov.uk/ukpga/2014/21/contents/enacted), 2014.

Wolpert, David H., and William G. Macready.: No Free Lunch Theorems for Optimization. IEEE Transactions on Evolutionary Computation, vol. 1, no. 1, pp. 67–82, doi:10.1109/4235.585893, **1997**.

Zymler, Steve, et al.: Worst-Case Value at Risk of Nonlinear Portfolios. Vol. 59, no. 1, pp. 172–88, **2013**.

APPENDIX

5   **Basic model.** Here we briefly describe the basic model presented by Muneepeerakul and Anderies (2017). The model shows dynamic behaviour of three principal variables, namely, the state of the public infrastructure, $I_{HM}$, resource level, $R$, and the fraction of time user makes use of infrastructure, $U$, through Eq's (A1, A4 and A5). The schematic diagram of this system of equations is shown in Figure A1.

Figure A1: Schematic diagram of the dynamical system model.  Taken from Muneepeerakul and Anderies (2017).

10   In this context, $I_{HM}$ depends on PIPs in term of maintenance cost and has a positive relationship with the capacity of users to create resource flows. Eq. (A1) illustrates the dynamics of $I_{HM}$ as follows:

$$\frac{dI_{HM}}{dt} = M(\dots) - \delta H(I_{HM}), \tag{A1}$$

where, $\delta$ is the infrastructure's depreciation rate and $H(I_{HM})$ states functional relationship of public infrastructure and productivity of each resource user. According to Muneepeerakul and Anderies (2017) many shared infrastructures

15   can be modelled by threshold functions. Given that $H(I_{HM})$ shows threshold behavior, they used a piecewise linear function to capture such behavior through Eq. (A2).

$$H(I_{HM}) = \begin{cases} 0, I_{HM} < I_0 \\ h\frac{I_{HM}-I_0}{I_m-I_0}, I_0 \leq I_{HM} \leq I_m, \\ h, I_{HM} \geq I_m \end{cases} \tag{A2}$$

where, $h$ represents maximum amount of harvest by each user under no restriction and $I_0$ and $I_m$ are lower bound and upper bound thresholds of $I_{HM}$ respectively. Also, $M(\dots)$ is maintenance function (Eq. A3) and depends on

20   social structure of the system.

$$M(\dots) = \mu_2 yCp RUN\cancel{U}H(I_{HM}), \tag{A3}$$

In Eq. (A3), given the number of users $N$, $RUN\cancel{NRU}H(I_{HM})$ is the total harvest from the natural infrastructure. The resource users sell total harvest at price $p$ to generate revenue. Subsequently, they assign a proportion $C$ of revenue to PIP's for their contribution. Meanwhile, the PIP's spend proportion $y$ of $C$ on maintaining public infrastructure

25   through the maintenance function $M(\dots)$. Also, $\mu_2$ is maintenance effectiveness of PIP's investment.

The second variable is resource level, $R$. They assumed the dynamics of resource to be:

$$\frac{dR}{dt} = G(R) - RUNH(I_{HM}) \tag{A4}$$

Natural infrastructure is assumed to invoke the conservation law comprising of regenerating capacity ($G(R) = g - dR$) and total unit of harvest, $RUN\text{NUR}H(I_{HM})$. The definition presented for $G$ is the simplest model for natural infrastructure where $g$ and $d$ are the natural replenishment and the loss rates, respectively.

The strategic behavior of the resource users ($RU's$) is captured by employing replicator equation. Indeed, replicator dynamics provide modeler with simple, realistic social mechanism where agents follow and replicate better-off strategies. The two possible strategies considered for RU's are staying inside system with the associated payoff of $\pi_U = (1-C)pRH(I_{HM})$ or leaving system with the payoff of $\pi_w = w$. According to replicator equation:

$$\frac{dU}{dt} = rU(1-U)(\pi_U - w) \qquad (A5)$$

Replicator equation discuss the fraction of time that RU's assign to working inside system given $C$ and $y$. Like RU's, there is also two alternatives for PIP's, working inside system or working for another CIS which leads to system failure. Meanwhile, $C$ and $y$ characterize the strategy or policy of PIPs. The PIPs will participate in this coupled system only when $\pi_p = (1-y)pCRUN\text{UNR}H(I_{HM}) \geq \pi_w$. In other words, the PIPs maintain in the system when they are better-off than working outside. This condition is termed the PIP Participation Constraint (PPC).

Based on the system of three differential equations (Eqs. A1, A4 and A5), the sustainable equilibria, i.e., long-term system outcomes that satisfy the stability condition and PIP Participation Constraint (PPC), can be expressed as follows:

$$i_{HM}^* = \frac{yCU^*NR^*}{g}H(I_{HM}^*); R^* = \frac{g}{d}\left(1 - \frac{i_{HM}^*}{yC}\right); U^* = \frac{(1-C)}{yC}\phi_1 i_{HM}^*, \qquad (A6)$$

where $i_{HM}^* := \frac{I_{HM}^*\delta}{\mu_2 pg}$ ( indicates dimensionless) and $\phi_1 = \frac{pg}{wN}$, a dimensionless group representing the relative lucrativeness of the system, namely the ratio of potential income—with the entire resource flow turned into income—relative to outside wage. The results reported in this study are based on the following parameter values:- $h = 0.0005$; $\delta = 0.1$; $I_0 = 0.3$; $I_m = 3$; $g = 100$; $d = 0.02$; $N = 1000$; $r = 0.15$; $p = 10$; $w = 1$; $w_p = 100$; $\mu_2 = 0.001$.

[Figure]

(a          (b)          (c

Figure 1: Resilience metrics for a specific setting (a $g$-$w$ combination) inside the sustainable region in the policy space (i.e., $C$-$y$ plane): (a) $R_{PPC}$ contours; (b) $R_{Stability}$ contours; and (c) $R_{system}$ contours. The black star in panels (a), (b), and (c) indicate the policy with the highest $R_{PPC}$, $R_{Stability}$, and $R_{system}$, respectively.

[Figure]

(a)                                           (b)

Figure 2: Variation of $R_{system}$ of a CIS with a fixed policy $(C, y)$ over 10,000 settings associated with uncertainty characterized by $\{g \in [75,125], w \in [0.75,1.25]\}$: (a) $R_{system}$ surface and (b) $R_{system}$ contours. The values of $R_{system}$ are used to calculate the mean, $\mu_{Rsystem}$, and the below-mean mean, $\mu_{R<\mu}$. In this particular case, the resilience does not change much when $g$ is greater than about 100 but becomes more sensitive to both $g$ and $w$ when $g$ is lower than 100.

[Figure]

(a)                                           (b)

[Figure]

| (c) | (d) |

Figure 3: the mean, $\mu_{R-system}$, and the below-mean mean of $R_{system}$, $\mu_{R<\mu}$, over entire decision space: (a) Surface of the resilience metric, $\mu_{R-system}$; (b) Contours of $\mu_{R-system}$; (c) Surface of the robustness, the below-mean mean ($\mu_{R<\mu}$); (d) Contours of the robustness, the below-mean mean ($\mu_{R<\mu}$)

[Figure]

Figure 4: Resilience-robustness trade-off. Each point represents, $\mu_{R-system}$ and $\mu_{R<\mu}$ of the coupled system with a given policy. The  blaeck dots represent a set of Pareto-optimal policies.

[Figure]

(a)                                                                    (b)

Figure 5: Pareto optimal policies, represented by  black dots  ──── in the policy space ($C$-$y$ plane), superimposed with resilience ($\mu_{R-system}$)  (a) and robustness ($\mu_{R<\mu}$) contours (b).

[Figure]

Figure A1: Schematic diagram of the dynamical system model. Taken from Muneepeerakul and Anderies (2017).